# Understanding the Learning Phases in Self-Supervised Learning via Critical Periods

**JangHyeon Lee**[1]    **Philipe Ambrozio Dias**[2]    **Yao-Yi Chiang**[1]    **Dalton Lunga**[2]
[1]University of Minnesota    [2]Oak Ridge National Laboratory

## Abstract

Self-supervised learning (SSL) has emerged as a powerful pretraining strategy to learn transferable representations from unlabeled data. Yet, it remains unclear how long SSL models should be pretrained to yield such representations. Contrary to the prevailing heuristic that longer pretraining translates to better downstream performance, we observe a *transferability trade-off*: across diverse SSL settings, intermediate checkpoints can yield stronger out-of-domain (OOD) generalization, whereas additional pretraining primarily benefits in-domain (ID) performance. From this observation, we hypothesize that SSL progresses through learning phases that can be characterized via the lens of critical periods (CP). Prior work on CP has shown that supervised models exhibit an early phase of high plasticity, followed by a consolidation phase where adaptability declines but task-specific performance increases. Since traditional CP analysis was developed for supervised settings, we rethink it for SSL in two ways. First, we inject deficits to perturb the pretraining data and assess their lasting impact on representation quality via downstream tasks. Second, we compute the Fisher Information on pretext objectives to track plasticity, quantifying how sensitive model parameters are to the pretext task. Our experiments suggest that SSL models may exhibit their own CP, with CP closure coinciding with a sweet spot for broad downstream transferability. Leveraging these insights, we introduce *CP-guided checkpoint selection* as a strategy for selecting checkpoints that offer stronger OOD transferability. Finally, to balance the transferability trade-off, we present *CP-guided self-distillation*, which selectively distills layer representations from the intermediate checkpoint into their overspecialized counterparts in the final checkpoint.

## 1 Introduction

Self-supervised learning (SSL) leverages pretext tasks (e.g., contrasting views or predicting masked inputs) to learn representations from unlabeled data that transfer well to downstream tasks (Liu et al., 2021; Ozbulak et al., 2023; Gui et al., 2024). While prior work has studied *how well* SSL models transfer (Newell & Deng, 2020; Ericsson et al., 2021a; Hu et al., 2021), *why* they transfer (Saunshi et al., 2019; Ericsson et al., 2021b), and *under what conditions* they succeed (Tian et al., 2020; Zhao et al., 2020; Cole et al., 2022; Dubois et al., 2022; 2023), it remains unclear *how long to pretrain SSL models* for transferable representations to emerge.

Without knowing when the SSL model has learned enough from its pretext task, pretraining risks both under- and over-training. Stopping too early yields underdeveloped representations, leading to the common heuristic that *longer pretraining is better* (Chen et al., 2020; He et al., 2022). However, pretraining for too long increases computational costs and risks overfitting to the pretext objective, potentially at the expense of broad downstream transferability.

Determining the optimal pretraining duration is difficult because SSL objectives are only implicitly aligned with downstream transferability (Balestriero et al., 2023; Reizinger et al., 2025). Typically, the quality of SSL representations is assessed *after pretraining* via linear probing or finetuning (Chen et al., 2020; Kumar et al., 2022; Balestriero et al., 2023). Such post-hoc evaluation is costly to repeat across tasks and, more importantly, provides no guidance *during pretraining* about whether learned representations are underdeveloped or already overspecialized to the pretext task.

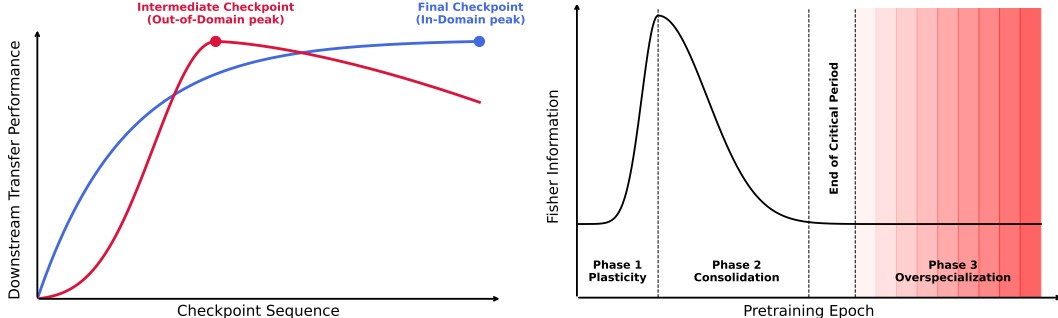

Figure 1: **(Left)** Conceptual schematic of downstream performance across pretrained checkpoints. In-domain (ID) performance increases with pretraining, whereas out-of-domain (OOD) transferability peaks at an intermediate checkpoint. **(Right)** Conceptual schematic of Fisher Information (FI) dynamics during SSL pretraining. The curve shows three phases. *Phase 1 (Plasticity)* shows a rise in FI when representations are highly sensitive to changes. *Phase 2 (Consolidation)* shows a decline in FI as representations stabilize. The critical period (CP) closes once FI stabilizes, before *Phase 3 (Overspecialization)*, where OOD performance declines. Red shading highlights the loss of transferability beyond CP closure. **(Takeaway)** CP closure may mark a sweet spot for broad transfer.

By evaluating checkpoints across the SSL pretraining trajectory (Fig. 1, left) we observe a *transferability trade-off*: intermediate checkpoints tend to achieve better out-of-domain (OOD) transfer than later checkpoints, whereas extended pretraining primarily benefits in-domain (ID) performance. Here, ID denotes finetuning on a labeled version of the pretraining dataset, while OOD denotes finetuning on labeled datasets drawn from different distributions (Marks et al., 2025). This transferability trade-off pattern implies that pretraining does not simply yield representations that improve uniformly across settings. Instead, we hypothesize that SSL progresses through *learning phases* whose representations differ in transferability: earlier phases may favor OOD generalization, while later phases specialize toward the pretraining distribution.

To build intuition, we draw on the notion of *critical periods* (CP) (Hensch, 2004). Prior work (Achille et al., 2018) shows that, much like biological systems, neural networks exhibit CP: an early window of high plasticity followed by a consolidation phase where representations stabilize and adaptability declines. In supervised learning, these phases were revealed through perturbation experiments, where temporary distortions of the training data permanently impaired generalization when applied during early epochs but had little effect later. This temporal sensitivity can be explained by Fisher Information (FI) (Fisher, 1925), which quantifies how strongly small parameter changes affect model predictions and serves as a proxy for *information plasticity* (Achille et al., 2018; Berariu et al., 2021). Early in training, FI rises and plasticity is high, so input perturbations strongly reshape representations and leave lasting effects. As training continues, FI declines and representations consolidate, so later perturbations have little impact. Overall, CP analyses reveal *when* representations are adaptable or rigid, which may offer insights into transferability.

Yet critical periods (CP) have not been studied in SSL, where transferability between pretraining and downstream tasks is key. Unlike supervised learning, SSL derives its supervisory signal from the data itself rather than explicit labels, making prior CP analyses not directly applicable. We therefore reformulate CP analyses to track information plasticity during SSL. This is achieved in two ways: (1) applying perturbations during pretraining to test stage-wise effects on downstream transferability, and (2) redefining Fisher Information (FI) with respect to pretext tasks.

We find that SSL pretraining may also exhibit a structured progression (Fig. 1, right). During pretraining, FI rises and then declines, marking a transition from plasticity to consolidation. We observe CP closure at the end of consolidation, where OOD performance tends to peak. Beyond CP closure, OOD generalization degrades, suggesting a phase of *overspecialization*. This pattern aligns with our empirical finding that intermediate SSL checkpoints can transfer better OOD than later ones.

Building on these observations, we explore whether critical periods (CP) can inform efficient and transferable SSL. We introduce two CP-guided strategies. *CP-guided checkpoint selection* uses CP closure to select checkpoints that favor OOD transfer, while pretraining beyond closure benefits ID

performance. To balance this trade-off, *CP-guided self-distillation* distills early-layer features from the CP closure checkpoint into the early layers of the final checkpoint during finetuning, while leaving later layers intact. We target early layers because they are widely understood to encode general features (Yosinski et al., 2014; Skean et al., 2025), which may erode with prolonged pretraining, while later layers encode task-specific information tied to the training objective, which in SSL is the pretext task (Bordes et al., 2022; 2023; Ouyang et al., 2025).

Our contributions can be summarized as follows:

- We observe a *transferability trade-off* in SSL pretraining. Intermediate checkpoints can yield stronger out-of-domain (OOD) transferability, while models pretrained longer tend to improve in-domain (ID) accuracy. This calls for rethinking the standard practice in SSL that longer pretraining translates to better representations for downstream tasks (§2).

- We connect this phenomenon to the notion of critical periods (CP), providing the *first study of CP in SSL* and their impact on transferability. Since SSL objectives differ from supervised settings, we reformulate CP analyses for SSL by introducing perturbations into pretraining and redefining Fisher Information in terms of pretext tasks. These analyses suggest that SSL models may exhibit their own CP (§3).

- We characterize an *overspecialization phase*, where prolonged pretraining reduces OOD generalization. Building on this insight, we introduce two interventions: *CP-guided checkpoint selection*, which uses CP closure to select checkpoints with stronger OOD transferability, and *CP-guided self-distillation*, which distills early-layer features from CP checkpoints into later checkpoints to recover OOD performance while retaining ID strength (§4).

## 2 DOES LONGER SELF-SUPERVISED PRETRAINING ALWAYS IMPROVE DOWNSTREAM TRANSFERABILITY?

Prior work in self-supervised learning (SSL) reported that longer pretraining improves downstream performance (Goyal et al., 2019; Chen et al., 2020; He et al., 2022). This has led to the de facto practice of pretraining SSL models for as long as compute budgets allow. We find that this improvement does not universally hold. Instead, we observe a **transferability trade-off**: *while extended pretraining improves in-domain (ID) performance, it can diminish out-of-domain (OOD) transferability.*

### 2.1 EXPERIMENTAL SETUP

To study how pretraining duration affects downstream transferability, we evaluate two families of SSL: discriminative and generative (Ozbulak et al., 2023). For discriminative SSL, we include a contrastive method, SimCLR (Chen et al., 2020), and non-contrastive methods, VICReg (Bardes et al., 2021) and DINO (Caron et al., 2021). For generative SSL, we use MAE (He et al., 2022). For architectures, we use a canonical ResNet (He et al., 2016) for SimCLR and VICReg, and a vision transformer (Dosovitskiy et al., 2020) for DINO and MAE. We pretrain on fMoW-RGB (Christie et al., 2018), a large-scale remote sensing dataset with well-defined distribution shifts (Koh et al., 2021; Rolf et al., 2024), making it a natural testbed for ID and OOD transfer.

We pretrain each model from scratch for 1000 epochs, saving checkpoints every 50 epochs. Downstream transfer is evaluated along two dimensions (Marks et al., 2025). In-domain (ID) performance is measured by finetuning and evaluating on a labeled version of the pretraining data. Out-of-domain (OOD) performance is measured on datasets whose distribution differs from the pretraining data. For each checkpoint, we finetune and compare against the model pretrained for 1000 epochs, which we refer to as the *final checkpoint*. Details on pretraining and downstream evaluation are in §A, and additional ImageNet (Deng et al., 2009) experiments in §B.

### 2.2 RESULTS

Fig. 2 shows downstream performance of SSL models across pretraining durations. OOD transfer peaks at intermediate checkpoints, while ID performance tends to increase. VICReg (Bardes et al., 2021), DINO (Caron et al., 2021), and MAE (He et al., 2022) all reach peak OOD performance at

intermediate checkpoints, while SimCLR (Chen et al., 2020) follows the same pattern but peaks considerably later. Although the exact timing varies, this divergence between OOD and ID is evident, suggesting that intermediate checkpoints can yield more broadly transferable representations.

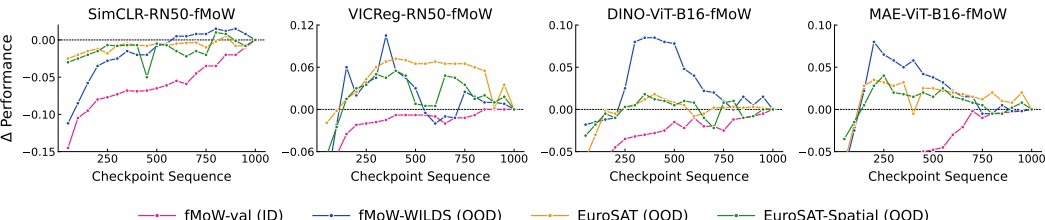

Figure 2: Transferability trade-off in SSL. The x-axis denotes pretraining epoch (checkpoints saved every 50 epochs), and the y-axis shows the change in performance relative to the final checkpoint.

## 3 CRITICAL PERIODS IN SELF-SUPERVISED LEARNING

Insights from §2 raise the question: *why do different stages of pretraining yield different transfer properties?* We hypothesize that SSL pretraining progresses through structured learning phases. To examine this, we draw on the notion of *critical periods* (CP) (Hensch, 2004). Prior work shows that neural networks exhibit phases of early plasticity, when representations are highly sensitive to change, followed by a consolidation phase with reduced plasticity (Achille et al., 2018; Kim et al., 2023). Yet whether such phases exist in SSL, and how they relate to transferability, remains unexplored. If SSL models pass through similar transitions, probing when plasticity is present or lost during pretraining may help explain the mechanism behind the observed transferability trade-off.

To investigate this, we revisit traditional critical period analyses in supervised learning (§3.1), followed by their reformulation for SSL via two approaches: perturbation experiments on unlabeled pretraining data (§3.2) and tracking Fisher Information on pretext tasks (§3.3).

### 3.1 PRIOR CRITICAL PERIOD ANALYSES REQUIRE RETHINKING FOR SSL

**How critical periods (CP) have been studied in supervised learning (SL).** Prior work identifies CP in two ways (Achille et al., 2018). First, perturbation experiments probe whether the *timing* of perturbations matters. If altering the input data distribution early in training degrades overall task performance, while the same change later has little effect, this marks a critical early phase. Second, Fisher Information (FI) (Fisher, 1925) analysis provides a continuous marker of plasticity. Put simply, FI quantifies how sensitive the model's predictions are to changes in its parameters (van de Ven, 2025). Achille et al. (2018) show that during training, FI rises then declines. Deficits introduced early, while FI is high, leave lasting effects because the model is actively forming its representations. Once FI declines, the model has consolidated and the same deficits have little effect.

**Why critical period analyses must be rethought for self-supervised learning (SSL).** The two probes above rely on supervised downstream signals. However, SSL pretraining is decoupled from downstream labels and optimizes proxy objectives on unlabeled data. One could study critical periods during finetuning, when downstream labels are available, but this only reveals how a *fixed representation* adapts to one task, not how transferable representations emerge *during pretraining*. Our focus is SSL pretraining itself, since this stage defines the representations that constrain all downstream learning. In particular, pretraining on unlabeled data $D_A$ can be viewed as producing a posterior $p(\theta \mid D_A)$ that serves as the prior for any downstream task $D_B$ (Shwartz-Ziv et al., 2022). To understand how this prior evolves, critical periods must be analyzed *during pretraining*, not after.

**Probing critical periods in SSL.** To study critical periods in SSL, we introduce two probes during pretraining. (1) *Deficit injection on the unlabeled pretraining data* perturbs the input distribution. The pretext task remains unchanged, but the self-supervision signal is degraded (e.g., input perturbations remove fine-grained cues, making data pairs harder to align or reconstructions less informative). By varying when deficits are introduced and measuring their impact on downstream transfer, we can identify phases when representations are more or less sensitive to change. (2) *Fisher Infor-*

*mation (FI) on pretext objectives* quantifies the sensitivity of model parameters to the supervisory signal defined by the pretext tasks. Tracking FI over pretraining reveals when parameters remain adaptable and when they consolidate, which is crucial in SSL since the value of pretraining lies in producing transferable representations. Identifying when representations are still malleable versus when they have stabilized can help explain when they are effective for downstream transfer.

### 3.2 PROBE 1: DEFICIT INJECTION ON PRETRAINING DATA

The central question is: *does the impact of input perturbations during SSL pretraining depend on* **when** *they occur?* If perturbations early in pretraining change the final representations, as reflected in downstream task performance, while the same perturbations later in pretraining have little effect, this would indicate a critical period in SSL.

Let $\mathcal{D} = \{x_i\}_{i=1}^N$ denote samples from a clean distribution $p(x)$. A model learns a representation function $f_\theta : \mathcal{X} \to \mathbb{R}^d$ with parameters $\theta$, trained with a self-supervised loss $\ell_{\text{SSL}}(f_\theta(x))$. To inject deficits, we replace clean training data with data drawn from a perturbed distribution $p'(x)$ starting at onset epoch $t_0$ and lasting for a duration of $\Delta t$ epochs. After this window, training resumes on clean data until epoch $T$, where $T > t_0 + \Delta t$.

We denote the encoder trained entirely on clean data as $f_\theta$ (baseline) and the encoder trained with a deficit window as $f_{\theta'}$. To quantify the effect of the intervention, we compare downstream task performance between these models. Let $\Phi(\cdot)$ denote a downstream evaluation metric (e.g., classification accuracy). The sensitivity score is defined as

$$S(t_0) = \Phi(f_\theta) - \Phi(f_{\theta'}). \tag{1}$$

This score reflects the relative degradation in downstream task performance caused by the perturbation. A critical period exists if early perturbations yield higher sensitivity than later ones.

**Deficit settings.** Following prior work, we simulate sensory deprivation by replacing inputs with Gaussian noise (Achille et al., 2018). For SSL methods, the pretext objectives (e.g., contrastive alignment (Chen et al., 2020) or masked reconstruction (He et al., 2022)) continue updating during the deficit window, but the supervisory signal now comes from noise rather than meaningful images. As a result, the model may learn nuisance features that are not useful for downstream transfer (Soatto & Chiuso, 2014; Achille & Soatto, 2018). Each deficit is applied for a fixed window (5, 30, 50 epochs) at varying onset times $t_0$: early (epoch 0), middle (epoch 450), and late (epoch 750), following (Kleinman et al., 2024). After the deficit window, training resumes on clean inputs (until epoch $T = 1000$). We use the same evaluation settings as in §2.1.

**Early SSL pretraining phases are sensitive.** Fig. 3 shows the sensitivity $S(t_0)$ of learned representations to noise deficits introduced at different times during pretraining. Across the evaluated SSL settings, we find that deficits applied at the start of pretraining lead to larger degradation than when the same deficits are introduced later. While the absolute magnitude of sensitivity varies by method, a similar pattern emerges: the beginning of pretraining appears to be a sensitive window where perturbations to the data distribution can leave lasting effects on learned representations.

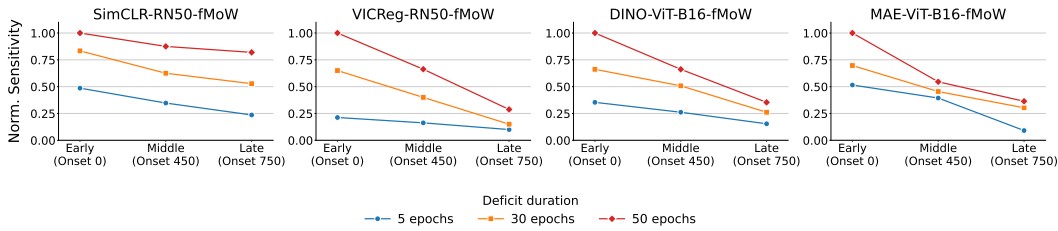

Figure 3: Sensitivity $S(t_0)$ to input perturbations introduced at different stages of SSL pretraining on fMoW. Checkpoints are then finetuned on fMoW-train and evaluated on held-out fMoW-val. Higher values indicate stronger lasting degradation in downstream accuracy relative to a clean baseline.

### 3.3 PROBE 2: TRACKING FISHER INFORMATION ON PRETEXT OBJECTIVES

Perturbation experiments reveal whether temporary interventions have lasting effects, but they do not explain *why* sensitivity varies across pretraining. To provide an analytical view, we study the evolution of Fisher Information (FI) (Fisher, 1925) during SSL pretraining. FI measures the sensitivity of the model's predictions to perturbations in its parameters and has been used to quantify parameter importance (Amari, 1998; Kirkpatrick et al., 2017). As a positive semi-definite approximation of the Hessian (Martens, 2020), FI also reflects the local curvature of the loss landscape (Lewandowski et al., 2023). Unlike the full Hessian, its trace can be estimated efficiently (Achille et al., 2018).

In SSL, pretext tasks define supervisory targets $y$ derived from the input $x$ (Balestriero et al., 2023; Balestriero & LeCun, 2024). For instance, in contrastive learning, $y$ specifies positive and negative pairs from augmentations of $x$, while in masked image modeling, $y$ denotes masked input regions to be reconstructed. More generally, these pretext objectives can be formulated as optimizing a conditional distribution $p_\theta(y|x)$ (Wang et al., 2024a; Bizeul et al., 2024; Alshammari et al., 2025). From this perspective, FI computed on $p_\theta(y|x)$ quantifies how sensitive parameters are to the pretext supervisory signal, and tracking this quantity over training reveals how that sensitivity evolves. Prior work has linked FI dynamics to plasticity (Achille et al., 2018; Berariu et al., 2021): FI rises as the model traverses high-curvature regions during early training, then declines as representations consolidate into flat minima. Since downstream tasks inherit these learned representations, plasticity dynamics can influence transferability. Specifically, models that lose plasticity risk settling into representations that are harder to adapt, while those that retain plasticity can readily adjust to new tasks (Achille et al., 2018; Lyle et al., 2023; Dohare et al., 2024; Springer et al., 2025).

Consider a model with parameters $\theta \in \mathbb{R}^d$, trained on inputs $x \sim \hat{p}(x)$ where $\hat{p}(x)$ is the empirical distribution of $\mathcal{D}$. To quantify local sensitivity, we consider an infinitesimal perturbation of the parameters, $\theta' = \theta + \delta\theta$. The effect of this perturbation is measured by the Kullback–Leibler (KL) divergence between $p_{\theta'}(y|x)$ and $p_\theta(y|x)$. A second-order Taylor expansion gives

$$\mathbb{E}_{x\sim\hat{p}(x)} \mathrm{KL}(p_\theta(y|x) \, \| \, p_{\theta'}(y|x)) = \tfrac{1}{2} \delta\theta^\top F \, \delta\theta \; + \; o(\|\delta\theta\|^2), \qquad (2)$$

where the Fisher Information Matrix (FIM) is

$$F := \mathbb{E}_{x\sim\hat{p}(x)} \mathbb{E}_{y\sim p_\theta(y|x)} \left[ \nabla_\theta \log p_\theta(y|x) \, \nabla_\theta \log p_\theta(y|x)^\top \right]. \qquad (3)$$

The matrix $F$ characterizes how perturbations to the parameters $\theta$ influence the model's predictive distribution. Parameter-space directions with large eigenvalues of $F$ correspond to high sensitivity, whereas directions with small eigenvalues can be altered with minimal impact on model behavior.

Since computing the full FIM is intractable, particularly when tracked across pretraining epochs, we use its trace as a scalar measure of total sensitivity (Achille et al., 2018; Jastrzebski et al., 2021):

$$\mathrm{tr}(F) = \mathbb{E}_{x\sim\hat{p}(x)} \mathbb{E}_{y\sim p_\theta(y|x)} \left[ \|\nabla_\theta \log p_\theta(y|x)\|^2 \right]. \qquad (4)$$

The trace of $F$ is the expected squared norm of the score function. In practice, since we compute this quantity at every epoch throughout pretraining, we approximate $\mathrm{tr}(F)$ using gradients of the self-supervised loss evaluated at observed targets from the pretext task. Despite known differences from the true Fisher (Kunstner et al., 2019), this approximation has been shown to effectively track trends in learning dynamics and parameter sensitivity (Lewandowski et al., 2023; Li et al., 2025).

**Plasticity rises, peaks, and stabilizes during SSL pretraining.** Fig. 4 shows Fisher Information (FI) trajectories during SSL pretraining, providing a quantitative view of how plasticity evolves over time. Across methods, FI rises early, peaks, and then declines before stabilizing.

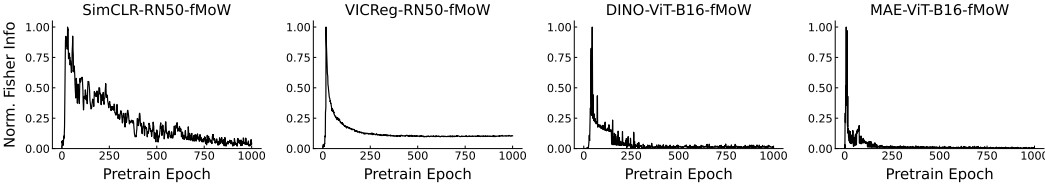

Figure 4: Fisher Information tracked during pretraining.

We define the *critical period* (CP) as the sequence of epochs before FI stabilizes. During this phase, the model is highly plastic and representations can be readily adapted to new information. Once FI stabilizes, the CP closes: the model discards variability uninformative for the pretext task, and representations become less sensitive to change. These dynamics align with the perturbation experiments in Fig. 3. Deficits introduced during the early phase had lasting effects on representation quality, whereas deficits introduced after the CP produced only minor effects, suggesting that the model had consolidated and become less responsive to perturbation. Overall, the rise and decline of FI may reflect an asymmetry in how SSL pretraining responds to new information over time.

## 4    CRITICAL PERIODS AS A GUIDE FOR EFFICIENT AND TRANSFERABLE SSL

In the previous section, our analyses revealed that SSL pretraining can exhibit its own critical period (CP) dynamics. Here, we investigate how these dynamics relate to downstream transferability and introduce two simple CP-guided interventions for efficient and transferable SSL.

### 4.1    ALIGNING CRITICAL PERIODS WITH DOWNSTREAM TRANSFERABILITY

To examine whether critical period (CP) dynamics relate to transferability trade-offs (§2), we align Fisher Information (FI) trajectories (Fig. 4) with downstream performance (Fig. 2) across pretraining epochs. Fig. 5 shows a recurring pattern across SSL methods: out-of-domain (OOD) transferability peaks near the point where FI stabilizes (grey shading marks CP closure), then declines even as in-domain (ID) accuracy continues to rise.

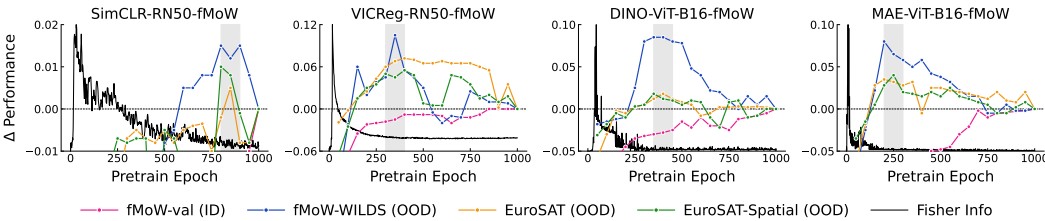

Figure 5: Relation between Fisher Information (FI) dynamics and downstream transferability. FI trajectories (black) are aligned with downstream performance (colored lines) across checkpoints.

**The overspecialization phase.** We define the divergence between rising ID and declining OOD performance as the onset of an *overspecialization phase*. After CP closure, representations continue to specialize on the pretext distribution by discarding variability irrelevant to the pretext task. While this pruning benefits ID performance, it also discards information that may be useful for OOD transfer, leading to a divergence between the two. This suggests that CP closure provides a *sweet spot* where representations are sufficiently learned but not yet overspecialized to the pretext task.

**Fisher Information (FI) dynamics help explain delayed trade-offs.** As noted in §2, SimCLR's transferability trade-off emerges later than in other SSL methods. FI trajectories suggest that Sim-CLR's critical period (CP) closes later, which may delay overspecialization. This aligns with prior findings that contrastive losses converge slowly (Shah et al., 2022; Tong et al., 2023). A key distinction is that SimCLR (Chen et al., 2020) is the only method among those we study whose objective depends on both positive and negative pairs. Since the gradients of the contrastive loss depend on each sample's position relative to all other negatives, the optimization objective shifts as the minibatch changes (Chen et al., 2022). This requires the model to repeatedly reorganize the global structure of the representation space to keep positives aligned while pushing all other samples apart. We conjecture that SimCLR's delayed CP-closure reflects this repeated global reshaping.

In contrast, VICReg (Bardes et al., 2021), DINO (Caron et al., 2021), and MAE (He et al., 2022) do not rely on negatives. VICReg regulates per-batch variance and covariance, DINO enforces alignment with a slowly updated teacher, and MAE reconstructs masked parts of a single image. These objectives do not require maintaining relationships to all other samples in the batch, unlike SimCLR (Chen et al., 2022), which can partly account for the faster CP closure we observe.

## 4.2 CRITICAL PERIOD-GUIDED CHECKPOINT SELECTION (CP-CS)

Selecting the optimal SSL checkpoint is non-trivial, as earlier checkpoints risk underdeveloped representations while later ones overspecialize to the pretext task. The finding that OOD transferability peaks near the end of the critical period (CP) suggests a practical strategy. Rather than defaulting to the conventional final checkpoint, we introduce *Critical Period-guided Checkpoint Selection (CP-CS)*, which leverages Fisher Information (FI) dynamics to identify CP closure checkpoints.

CP-CS requires no additional cost beyond pretraining and no labels. One can (i) monitor FI trace across epochs, (ii) identify the range of epochs where the FI curve stabilizes, and (iii) select the nearest checkpoint for downstream. This rule-of-thumb can narrow the search space: CP closure provides a reasonable starting point when OOD transfer is important, while pretraining beyond CP closure remains beneficial when ID accuracy is the priority. Additional results are provided in §C.

## 4.3 CRITICAL PERIOD-GUIDED SELF-DISTILLATION (CP-SD)

While intermediate checkpoints exhibit stronger out-of-domain (OOD) generalization, later checkpoints continue to achieve higher in-domain (ID) accuracy. This trade-off reflects complementary properties: *CP checkpoints* can capture broadly transferable features, while *post-CP checkpoints* specialize toward pretext-specific signals, increasing alignment with the pretraining distribution.

To mitigate this loss of OOD transferability, we present *CP-guided self-distillation* (CP-SD), a light post-pretraining strategy that reuses existing checkpoints. The idea is simple: use the CP checkpoint as a teacher for the intermediate layers of the post-CP checkpoint (student). During downstream finetuning, we optimize the task loss $L_{\text{task}}$ (e.g., cross-entropy for classification) together with a distillation loss applied only to intermediate layers $\mathcal{L}$. The overall objective is

$$L = L_{\text{task}} + \lambda \sum_{l \in \mathcal{L}} \|f_l^{\text{student}} - f_l^{\text{teacher}}\|_2^2, \tag{5}$$

where $\lambda$ is a hyperparameter and the later layers are optimized only with $L_{\text{task}}$.

**The intuition** comes from our layer-wise probing analysis (Fig. 6): CP checkpoints achieve stronger OOD performance across the network, with the gap largest in the early layers. In contrast, post-CP checkpoints provide higher ID accuracy in the later layers, reflecting the benefits of extended pretraining when downstream tasks are aligned with the pretraining distribution. While early layers are widely understood to encode general features (Yosinski et al., 2014), this property can weaken throughout training. As pretraining progresses, the model may increasingly compress representations toward information relevant to the task-specific objective (Shwartz-Ziv & Tishby, 2017), including in the early layers. CPSD addresses this trade-off by restoring the early layers of the final checkpoint toward their CP state to recover OOD transferability, while preserving the later layers of the post-CP checkpoint to maintain the ID strength gained through extended pretraining.

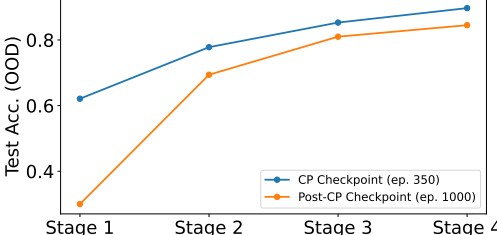 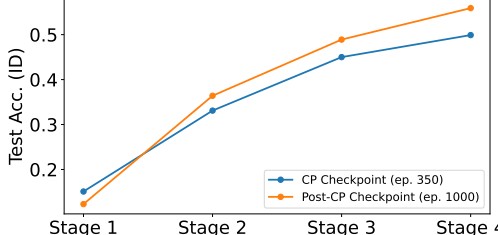

Figure 6: Layer-wise probing for OOD transfer (left: VICReg pretrained on fMoW, evaluated on EuroSAT-Spatial) and for ID performance (right: VICReg pretrained and evaluated on fMoW).

**Results.** Tab. 1 reports top-1 downstream classification accuracy. The final pretrained checkpoint achieves the strongest ID performance but suffers OOD degradation. The CP-guided checkpoint (CP-CS) shows the opposite pattern: it trades a small amount of ID accuracy for a large OOD gain. CP-guided self-distillation (CP-SD) aims to combine both properties: distilling early-layer features from the CP checkpoint into the final checkpoint yields a balanced overall performance. Distilling

all layers, however, performs worse than early-layer distillation. We suspect this is because pulling the entire model toward the CP checkpoint restores some generality from the intermediate model but also overwrites the useful ID-specific refinements learned in later layers.

Table 1: Downstream classification results for VICReg-RN50-fMoW. Results are averaged over 3 runs. **Bold**: best, underline: second best. Distillation settings are provided in §D.

| Model | fMoW-val (ID) | fMoW-WILDS (OOD) | EuroSAT (OOD) | EuroSAT-Spatial (OOD) |
|---|---|---|---|---|
| Final checkpoint | **0.621** $\pm$ 0.021 | 0.341 $\pm$ 0.034 | 0.917 $\pm$ 0.017 | 0.894 $\pm$ 0.028 |
| CP checkpoint | 0.610 $\pm$ 0.025 | 0.430 $\pm$ 0.031 | 0.931 $\pm$ 0.013 | 0.912 $\pm$ 0.022 |
| CP-SD (early layers) | 0.617 $\pm$ 0.018 | **0.445** $\pm$ 0.029 | **0.944** $\pm$ 0.011 | **0.925** $\pm$ 0.019 |
| CP-SD (all layers) | 0.611 $\pm$ 0.019 | 0.421 $\pm$ 0.023 | 0.929 $\pm$ 0.049 | 0.908 $\pm$ 0.012 |

## 5 DISCUSSION & RELATED WORK

In this work, we studied a simple yet underexplored question: *how long should we pretrain self-supervised learning (SSL) models?* Contrary to the prevailing heuristic that longer pretraining translates to better downstream performance (Chen et al., 2020; He et al., 2022), we find that the answer is more nuanced. Specifically, earlier checkpoints achieve stronger out-of-domain (OOD) transfer than later ones, while the latter improve in-domain (ID) performance. The transferability trade-off across pretraining duration suggests that SSL may undergo a phase transition, akin to *critical periods*.

**Critical early learning phases.** Originating in biology, critical periods refer to windows of heightened plasticity during which neural circuits are particularly sensitive to early experience (Kandel et al., 2000; Hensch, 2004; Knudsen, 2004). A similar effect has been reported in artificial neural networks: changes in the early training phase shape the final representation, whereas changes later have limited impact. Input perturbations applied early permanently reduce generalization, while the same perturbations applied later are recoverable (Achille et al., 2018; Kleinman et al., 2023; 2024; Altıntaş et al., 2025). Moreover, regularization methods (e.g., weight decay) only have large effects when applied early in training (Golatkar et al., 2019; Liu et al., 2020; Kalra & Barkeshli, 2023). Conceptually, critical periods mark a transition from a high-plasticity stage, where representations are rapidly formed, to a consolidation stage, where representations stabilize and task-irrelevant information is discarded (Shwartz-Ziv & Tishby, 2017; Achille et al., 2018; Zhou et al., 2025).

**Exploring critical periods in SSL.** Whether SSL exhibits critical periods (CP) similar to supervised learning, and how these phases affect downstream transfer, remains unexplored. Building on recent calls for a temporal perspective on SSL (Simon et al., 2023; Reizinger et al., 2025), we investigate the emergence of CP in SSL and their impact on transferability. Our results suggest that SSL pretraining undergoes structured phases: early epochs exhibit high plasticity, while later epochs consolidate the model into patterns dictated by the pretraining setup. Beyond plasticity and consolidation, we identify a subsequent phase of *overspecialization*. During overspecialization, OOD generalization declines, suggesting that representations become increasingly bound to pretraining data and pretext task. These dynamics shed light on when representations are broadly transferable, complementing prior work that investigated SSL transferability *after* full pretraining (Ericsson et al., 2021a;b).

**Several implications follow.** SSL is often positioned as a pathway to task-agnostic representations (Qiang et al., 2024; Reizinger et al., 2025). This has fueled the rise of foundation models, whose general-purpose representations transfer across tasks (Bommasani, 2021). From this view, SSL pretraining defines a distribution over parameters that serves as a prior for all possible downstream tasks. This prior is only useful to the extent that it supports adaptation, yet SSL is not devoid of specialization. Even without labels, every pretext objective imposes implicit supervisory signals (Balestriero & LeCun, 2024; Wang et al., 2024b), shaping the invariances and biases the model encodes. Since downstream tasks are unknown at pretraining time, SSL has no guidance for distinguishing task-relevant from task-irrelevant variation (Kleinman et al., 2021). Thus, models may capture nuisances alongside useful features (Xiao et al., 2020; Robinson et al., 2021; Wang et al., 2022; Bandara et al., 2023; Rabin et al., 2024; Qiang et al., 2025), and with extended pretraining, the prior may increasingly align with the pretext task, potentially reducing broad transferability. This tension is salient for foundation models, whose utility depends on SSL producing broadly adaptable priors. We briefly discuss this tension through an information-theoretic lens in §E.

## 6 LIMITATIONS & FUTURE WORK

Our analysis centers on unimodal vision models and a subset of SSL methods. Whether similar phase-like behavior occurs in other datasets and domains (e.g., language, time-series, audio) or in other SSL settings (e.g., joint embedding predictive architectures (Assran et al., 2023), multimodal SSL (Zong et al., 2024)) remains an open question. Additionally, the quality of the SSL representations is shaped by the interplay between training configurations such as augmentation choices in discriminative methods or masking logic in generative methods (Cabannes et al., 2023). Varying these configurations may uncover further insights into the dynamics of SSL (Reizinger et al., 2025).

Our experiments also operate within a fixed epoch budget, which may preclude observing phenomena such as epoch-wise double descent (Nakkiran et al., 2021; Stephenson & Lee, 2021), where test performance can worsen and then improve again with prolonged training. This is particularly relevant as SSL methods are known to be rather *inefficient* learners (Wang et al., 2021; Tong et al., 2023), and extending pretraining may reveal additional non-monotonic dynamics whose shape could vary across downstream settings.

Another promising direction is integrating label-free representation quality metrics such as $\alpha$-ReQ (Agrawal et al., 2022), RankMe (Garrido et al., 2023), and LiDAR (Thilak et al., 2023) into critical period analysis. Since no single metric fully captures transferability (Agostinelli et al., 2022), combining complementary metrics may provide a richer view: $\alpha$-ReQ characterizes the eigenspectrum decay of the feature covariance, RankMe measures its effective rank, and LiDAR estimates linear probing performance by measuring alignment between learned features and discriminative directions inferred from augmentations. Tracking such metrics *during pretraining* may offer a comprehensive picture of *when* structured representations begin to form and how they evolve temporally.

While our exploration is by no means exhaustive, our findings show that transferability in SSL can follow a non-monotonic trajectory. Analyses that rely solely on the final pretrained checkpoint therefore risk missing the stages at which broad transferability is acquired, altered, or lost. Understanding these transient phases may be as important as evaluating the converged model, as the final checkpoint is not necessarily the most transferable one.

## ACKNOWLEDGMENTS

This manuscript has been authored by UT-Battelle, LLC under Contract No. DE-AC05-00OR22725 with the U.S. Department of Energy. The United States Government retains and the publisher, by accepting the article for publication, acknowledges that the United States Government retains a non-exclusive, paid-up, irrevocable, world-wide license to publish or reproduce the published form of this manuscript, or allow others to do so, for United States Government purposes. The Department of Energy will provide public access to these results of federally sponsored research in accordance with the DOE Public Access Plan (http://energy.gov/downloads/doe-public-access-plan). This research used resources of the Oak Ridge Leadership Computing Facility at the Oak Ridge National Laboratory, which is supported by the Office of Science of the U.S. Department of Energy under Contract No. DE-AC05-00OR22725. The results and models presented in this work used compute resources from the National AI Research Resource Pilot (NAIRR), with support from NVIDIA, including NVIDIA's DGX Cloud product and the NVIDIA AI Enterprise Software Platform.

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

# A EXPERIMENTAL DETAILS (§2.1)

All experiments were run on NAIRR using NVIDIA A100-SXM4-80GB GPUs with mixed-precision training.

## A.1 PRETRAINING SETUP

**Hyperparameters.** For SimCLR (Chen et al., 2020), we use a batch size of 1024 with SGD, momentum 0.9, base learning rate 0.3, and weight decay $1 \times 10^{-4}$, with a 10-epoch linear warmup followed by cosine learning rate decay. The contrastive NT-Xent loss uses temperature $\tau = 0.1$ with negatives gathered across all GPUs. The projection head is a 2-layer MLP ($2048 \rightarrow 2048 \rightarrow 128$) with batch normalization and ReLU after the hidden layer. For VICReg (Bardes et al., 2021), we use LARS with momentum 0.9, weight decay $1 \times 10^{-6}$, and base learning rate 0.3 with linear scaling, with a 10-epoch linear warmup followed by cosine learning rate decay. The loss uses invariance, variance, and covariance weights $(\lambda, \mu, \nu) = (25, 25, 1)$. The projection head is a 2-layer MLP ($2048 \rightarrow 8192 \rightarrow 8192$) with batch normalization and ReLU after the hidden layer. For DINO (Caron et al., 2021), we use AdamW with base learning rate $5 \times 10^{-4}$ and linear scaling, with a 10-epoch linear warmup followed by cosine learning rate decay. Weight decay is cosine-scheduled from 0.04 to 0.4. The student temperature is $\tau_s = 0.1$ and the teacher temperature is fixed at $\tau_t = 0.04$. Teacher momentum increases from 0.996 to 1.0 following a cosine schedule, with center momentum 0.9. The student backbone uses stochastic depth with drop path rate 0.1. The projection head is a 3-layer MLP ($768 \rightarrow 2048 \rightarrow 2048 \rightarrow 256$) with GELU activations, followed by $\ell_2$ normalization and a weight-normalized linear layer projecting to 65,536 dimensions. The last layer of the student head is frozen for the first epoch. For MAE (He et al., 2022), we use AdamW with betas $(0.9, 0.95)$, base learning rate $1.5 \times 10^{-4}$ with linear scaling, and weight decay 0.05, with a 40-epoch linear warmup followed by cosine learning rate decay. The masking ratio is 0.75 and the reconstruction target is raw pixel values with mean squared error loss. The decoder has 8 Transformer blocks with embedding dimension 512, 16 attention heads, and MLP ratio 4.0.

**Augmentations.** For SimCLR (Chen et al., 2020), we apply random resized crop to $224 \times 224$ pixels (scale 0.08–1.0), random horizontal flip ($p = 0.5$), color jitter (brightness, contrast, saturation = 0.8, hue = 0.2; applied with $p = 0.8$), random grayscale ($p = 0.2$), and Gaussian blur ($\sigma \in [0.1, 2.0]$; $p = 0.5$). For VICReg (Bardes et al., 2021), we use the same pipeline as SimCLR but with reduced color jitter strength (brightness = 0.4, contrast = 0.4, saturation = 0.2, hue = 0.1; applied with $p = 0.8$) and a random solarization ($p = 0.1$). For DINO (Caron et al., 2021), we use multi-crop with two global views ($224 \times 224$ pixels, scale 0.4–1.0) and eight local views ($96 \times 96$ pixels, scale 0.05–0.4). All views use random horizontal flip ($p = 0.5$), color jitter (brightness = 0.4, contrast = 0.4, saturation = 0.2, hue = 0.1; applied with $p = 0.8$), random grayscale ($p = 0.2$), and Gaussian blur ($\sigma \in [0.1, 2.0]$; $p = 1.0$ for the first global view, $p = 0.1$ for the second, $p = 0.5$ for local views). Solarization ($p = 0.2$) is applied only to the second global view. For MAE (He et al., 2022), we apply random resized crop to $224 \times 224$ pixels (scale 0.2–1.0) and random horizontal flip ($p = 0.5$).

## A.2 DOWNSTREAM SETUP

**Datasets.** We follow the definition of in-domain (ID) and out-of-domain (OOD) transfer from Marks et al. (2025). For models pretrained on fMoW-RGB, ID transfer is measured on the held-out fMoW-RGB validation split. We consider three OOD datasets. fMoW-WILDS (Koh et al., 2021) introduces temporal and regional distribution shifts over the same categories. EuroSAT (Helber et al., 2019) contains 27,000 Sentinel-2 satellite images across 10 land-use classes. We use the RGB version at $64 \times 64$ pixels, which differs from fMoW-RGB in sensor, spatial resolution, and scene semantics. EuroSAT-Spatial (Stewart et al., 2022) uses the same EuroSAT images but splits train, validation, and test sets by longitude (60/20/20) to induce a spatial distribution shift. For models pretrained on ImageNet, ID transfer is measured on the held-out ImageNet validation split. We consider three OOD datasets with increasing domain shift: SUN397 (Xiao et al., 2010) for scene recognition within the natural image domain, EuroSAT-Spatial (Stewart et al., 2022) for remote sensing transfer, and Camelyon17-WILDS (Koh et al., 2021) for histopathological tumor detection.

**Evaluation.** We follow standard evaluation protocols in SSL (Balestriero et al., 2023). All models are evaluated using top-1 accuracy via finetuning. We use official splits when available and adopt a standard 80/20 split otherwise. For ResNet backbones (SimCLR, VICReg), we finetune for 50 epochs using SGD with momentum 0.9 and a base learning rate 0.1 with linear scaling. For ViT backbones (DINO, MAE), we finetune for 50 epochs using AdamW with betas $(0.9, 0.999)$ and a base learning rate $1 \times 10^{-3}$ with linear scaling.

# B   ADDITIONAL: IMAGENET RESULTS

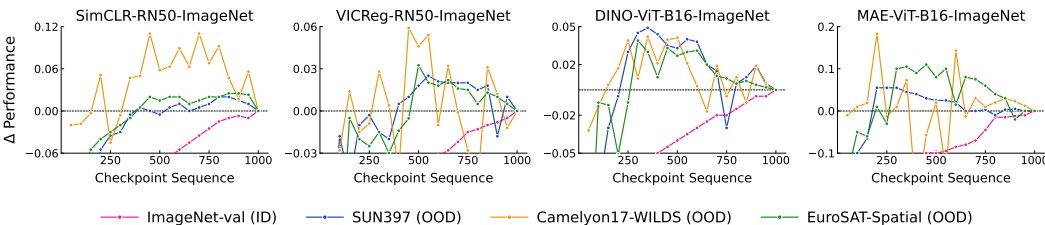

Figure 7: Transferability trade-off in SSL. The x-axis denotes pretraining epoch (checkpoints saved every 50 epochs), and the y-axis shows the change in performance relative to the final checkpoint.

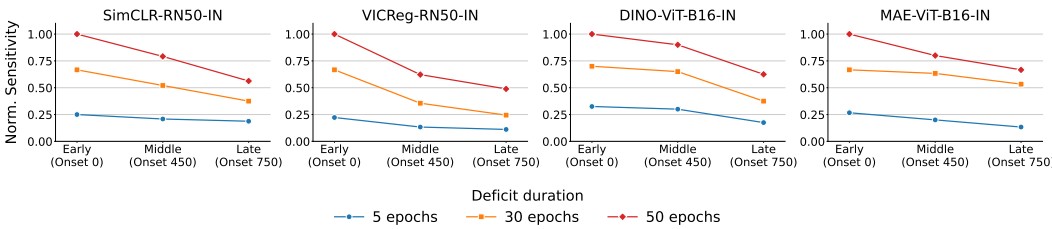

Figure 8: Sensitivity to input perturbations during pretraining reflected in downstream performance.

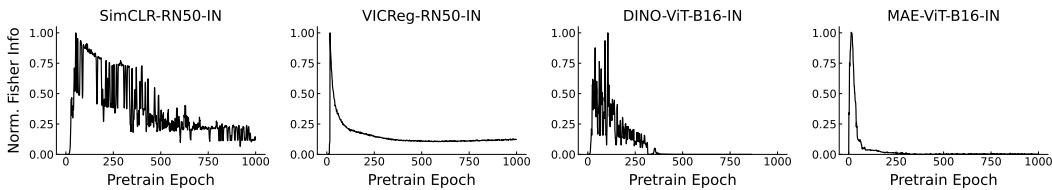

Figure 9: Fisher Information tracked across pretraining epochs.

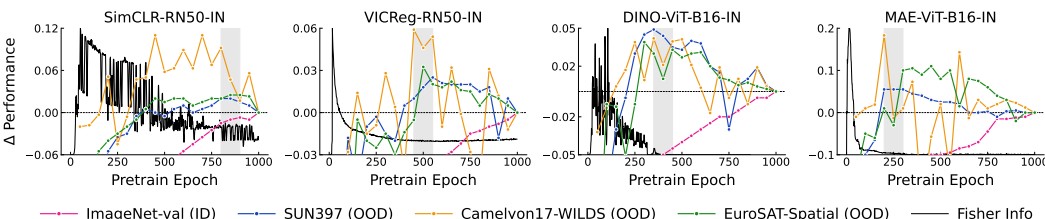

Figure 10: Relation between Fisher Information (FI) dynamics and downstream transferability. FI trajectories (black) are aligned with downstream performance (colored lines) across checkpoints.

## C  ADDITIONAL: CP-GUIDED CHECKPOINT SELECTION (CP-CS)

**Additional SSL method.** We pretrain DINOv2 (Oquab et al., 2023) with an EMA teacher and a student trained using AdamW with a cosine warmup schedule. Weight decay is annealed from $0.04$ to $0.4$ with a cosine schedule. The loss combines an image-level DINO objective (class tokens), a patch-level iBOT objective (masked tokens, applied only on the student), and a KoLeo regularizer ($\lambda = 0.1$). Teacher momentum is scheduled from $0.992$ to $1.0$. We use a ViT-S/16 backbone with DropPath $0.1$, LayerScale $10^{-5}$, and standard DINO multi-crop augmentations. For evaluation, we follow §A.2.

**Results.** Fig. 11 shows the Fisher Information (FI) dynamics during DINOv2 pretraining and their relation to downstream transfer. Checkpoints around FI stabilization coincide with peak OOD performance.

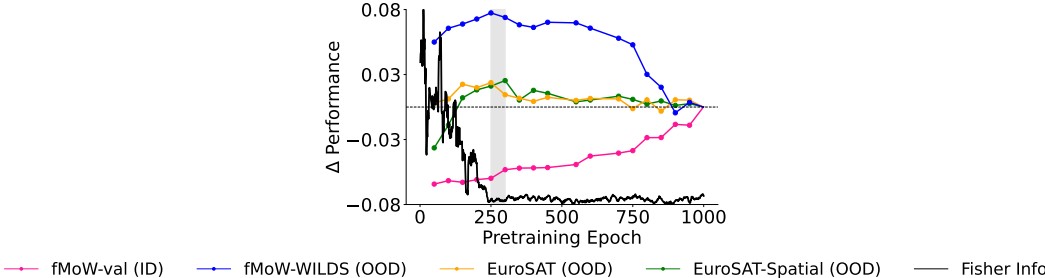

Figure 11: Relation between Fisher dynamics and transferability for DINOv2-ViT-S16-fMoW.

## D  ADDITIONAL: CP-GUIDED SELF-DISTILLATION (CP-SD)

**Distillation setting.** For distillation, we use the *first residual stage* (`layer1`) of ResNet-50 (He et al., 2016) and the first three Transformer blocks of ViT-B16 (aligning [CLS] tokens). Models are trained with AdamW for 100 epochs (batch 256, base LR $10^{-4}$ with cosine schedule, weight decay 0.05), using $\lambda = 0.5$ (Eq. 5).

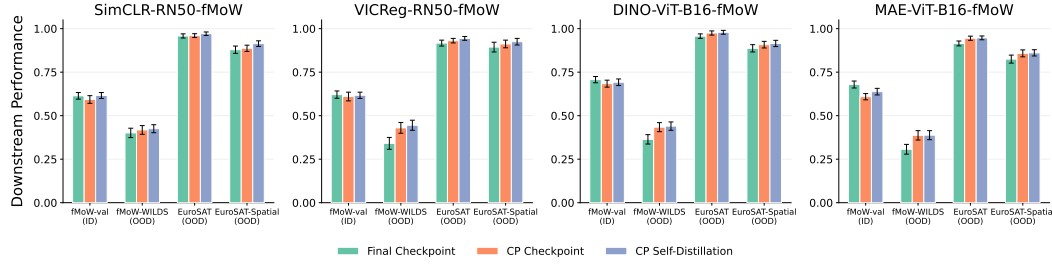

Figure 12: We compare the final checkpoint, the CP checkpoint, and our CP self-distilled (CP-SD) checkpoint. CP-SD consistently achieves the highest OOD performance while recovering ID.

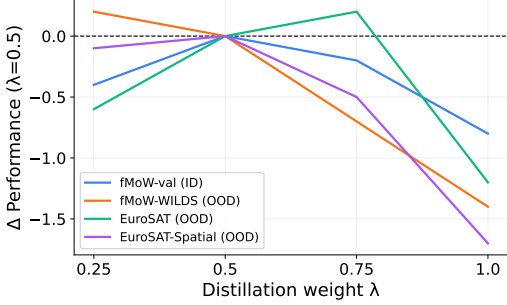

Figure 13: We use VICReg-RN50-fMoW and distill from the conv2_x block group (He et al., 2016). Performance is overall stable, though larger values ($\lambda = 1.0$) show diminishing returns.

# E   AN INFORMATION-THEORETIC LENS ON SSL SPECIALIZATION

**Background.** The Information Bottleneck (IB) (Tishby et al., 2000) provides a framework for characterizing optimal representations. Given an input $X$ and a task variable $Y$, the IB characterizes the representation $Z$ that optimally balances compression of $X$ against retention of information about $Y$:

$$\min_{p(z|x)} I(X;Z) - \beta I(Z;Y), \tag{6}$$

where $\beta$ controls the trade-off between the two terms. Prior work on the information dynamics of deep networks (Shwartz-Ziv & Tishby, 2017) shows that supervised training can proceed in two phases: a *fitting phase* in which $I(Z;Y)$ increases rapidly, followed by a *compression phase* in which $I(X;Z)$ decreases, though the generality of this pattern depends on architecture and estimation choices (Saxe et al., 2019).

Although self-supervised learning (SSL) does not explicitly optimize this IB objective, the IB perspective offers an intuition that may help explain the learning dynamics we observe. *Our goal here is not to claim a formal equivalence, but to use the IB framework as a preliminary interpretive lens.*

**In SSL**, downstream labels are unavailable during pretraining. Instead, each method defines a pretext signal $Y_{\text{pretext}}$, such as the positive-pair structure in contrastive learning (Chen et al., 2020), teacher predictions in self-distillation (Caron et al., 2021), or masked regions to reconstruct (He et al., 2022). Under the IB perspective, optimization encourages the representation to increase $I(Z; Y_{\text{pretext}})$ while discarding input variability that does not help predict $Y_{\text{pretext}}$, reducing $I(X; Z)$.

As optimization progresses, the representation specializes toward what $Y_{\text{pretext}}$ treats as relevant. Since the pretext signal is derived from the data itself, this relevance inherits the statistics of the pretraining distribution. Features irrelevant to $Y_{\text{pretext}}$ may be discarded even when they would be broadly useful, while distribution-specific correlations that happen to be predictive of $Y_{\text{pretext}}$ may be retained even when they do not generalize.

**In discriminative methods**, $Y_{\text{pretext}}$ encodes which augmented views originate from the same input. Recent unifying work (Alshammari et al., 2025) shows that SimCLR (Chen et al., 2020) and VICReg (Bardes et al., 2021) minimize the same divergence between a supervisory distribution over augmented views and a learned distribution over embeddings, and DINO (Caron et al., 2021) applies similar pressure through teacher-student alignment. Since the augmentations are applied to the pretraining distribution, the invariances the encoder learns are shaped by the statistics of that data. The pretext objective has no mechanism to distinguish between spurious and generalizable correlations: both are encoded so long as they satisfy the view-matching criterion.

**In generative methods**, $Y_{\text{pretext}}$ is defined by masked-region reconstruction. The decoder reconstructs individual patches from nearby tokens using a pixel-level loss, so the encoder is primarily rewarded for features that support local spatial prediction. Recent work formalizing MAE through the IB (Huang et al., 2025) provides further evidence: the reconstruction objective can produce a suboptimal compression trade-off, retaining more low-level detail than necessary while failing to preserve cross-view information that proves useful downstream.

**Takeaway.** The pretext task and the pretraining distribution jointly determine what counts as relevant information in SSL. When the downstream distribution diverges sufficiently, features shaped by this definition can impair performance (Wang et al., 2019; Teney et al., 2023; Deng et al., 2023). This does not imply that longer pretraining is strictly harmful: when pretext and downstream objectives are aligned, additional training may continue to improve useful features. Overall, extended pretraining may increasingly specialize the representation toward pretext-defined relevance, regardless of whether those features are transferable.

# F ABLATION STUDY

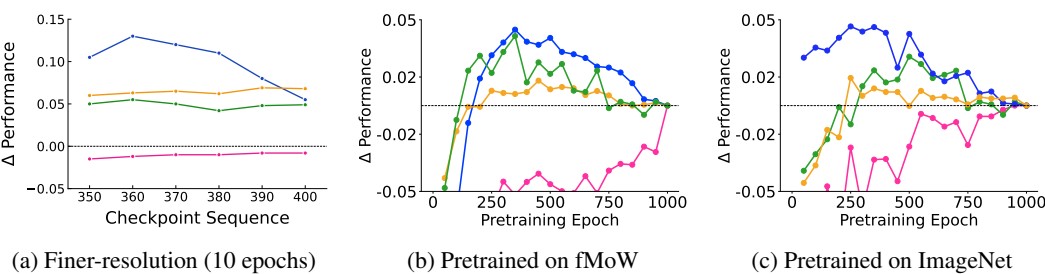

(a) Finer-resolution (10 epochs)    (b) Pretrained on fMoW    (c) Pretrained on ImageNet

Figure 14: **(a)** Evaluating every 10 epochs within the CP window for VICReg-RN50-fMoW: OOD performance (other than pink) peaks again nearby. **(b, c)** Linear probing across pretraining checkpoints: intermediate checkpoints outperform the final checkpoint on OOD tasks (other than pink), while ID performance (pink) tends to improve.

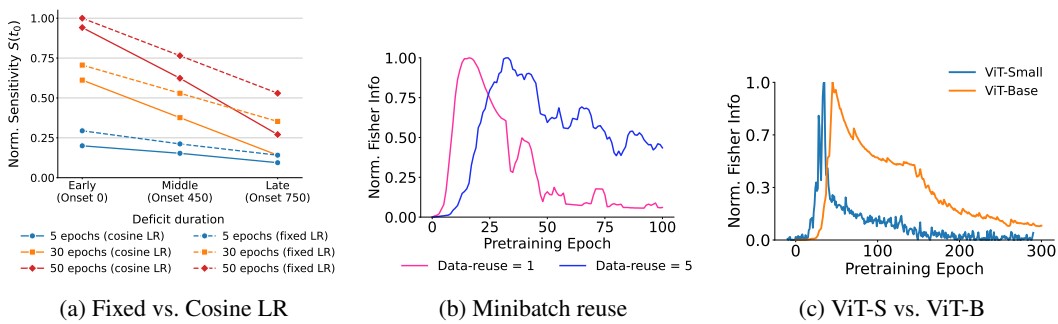

(a) Fixed vs. Cosine LR    (b) Minibatch reuse    (c) ViT-S vs. ViT-B

Figure 15: **(a)** Under both fixed and cosine LR schedules, early deficits cause the largest degradation, suggesting the temporal sensitivity is not necessarily an artifact of the LR schedule. **(b)** Increasing minibatch reuse from $k=1$ to $k=5$ causes FI to peak earlier, consistent with reduced gradient diversity accelerating CP closure. **(c)** ViT-Base shows slower FI stabilization than ViT-Small under DINO pretraining, suggesting larger capacity can delay CP closure.

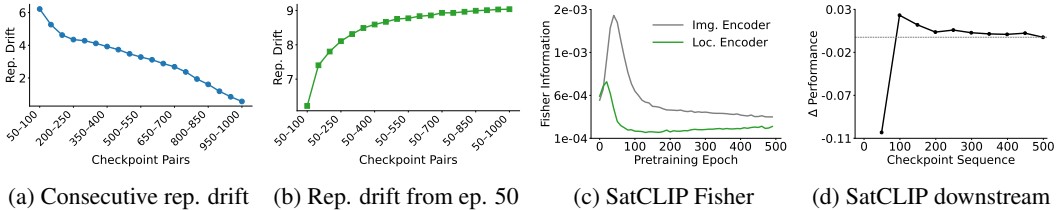

(a) Consecutive rep. drift    (b) Rep. drift from ep. 50    (c) SatCLIP Fisher    (d) SatCLIP downstream

Figure 16: **(a)** Representation drift for VICReg-RN50-IN: consecutive checkpoint drift is large early and decreases, **(b)** while drift relative to epoch 50 rises then plateaus, suggesting early pretraining exhibits greater representational plasticity. **(c)** Extending CP experiments to multimodal SatCLIP (Klemmer et al., 2025): FI peaks early and stabilizes for both encoders during pretraining, **(d)** and the intermediate checkpoint outperforms the final checkpoint across seven geospatial tasks.

