# OpenReview forum: "Understanding the Learning Phases in Self-Supervised Learning via Critical Periods"
_ICLR.cc/2026/Conference — ICLR 2026 Poster_

### Official Review · Reviewer_EgnN · 2025-10-27

**Soundness:** 3
**Presentation:** 3
**Contribution:** 3
**Rating:** 6
**Confidence:** 3

**Summary:**

This paper investigates the temporal dynamics of self-supervised learning (SSL) and its effect on transferability. The authors identify a transferability trade-off: intermediate checkpoints during SSL pretraining often yield stronger out-of-domain (OOD) generalization, whereas longer training enhances in-domain (ID) accuracy. Authors characterize SSL pretraining into three phases—plasticity, consolidation, and overspecialization—tracked via Fisher Information dynamics and deficit injection experiments. They further propose 2 training scheme for improvement:
1. CP-guided Checkpoint Selection (CPCS) for identifying checkpoints near CP closure, improving OOD transfer;
2. CP-guided Self-Distillation (CPSD) to distill representations from CP checkpoints into later ones, mitigating overspecialization effects

**Strengths:**

1. Proposed that the learning process of SSL is stage-by-stage and provide evidences for verification.
2. Proposed 2 training improvement scheme CPCS and CPSD without uesag of extra label.
3. Propose novel explanation for OOD generalization decay at the end of training.

**Weaknesses:**

1. The paper defines Critical Period closure as the epoch when the Fisher Information (FI) curve stabilizes—operationalized as a near-zero slope. It better comes up with a formal metric consider many other variables like batch size, hyperparameters and even downstream tasks.
2. Need to provide more insights about why the FI dynamic performance differently among different SSL architectures. Are these differences come from the loss term or the contrastive and reconstruction-based SSL methods？

**Questions:**

1. Can Fisher Information be replaced with a more stable unsupervised proxy?
2. Do critical periods also exist in language or multi-modal SSL?
3. How the model capacity will influence the CP closure?

---

> ### Author Response · Authors · 2025-11-28
>
> > **W1 The paper defines Critical Period closure as the epoch when the Fisher Information (FI) curve stabilizes—operationalized as a near-zero slope. It better comes up with a formal metric consider many other variables like batch size, hyperparameters and even downstream tasks.**
>
> We agree that defining critical period (CP) closure solely through stabilization of the Fisher Information (FI) curve is only one possible operationalization, and that a more formal metric could in principle incorporate factors such as batch size, learning rate schedule, or downstream-task behavior. We view our formulation as one first step toward such a broader metric.
>
> FI is useful in our SSL setting because it provides a qualitative indicator of “information plasticity,” meaning the extent to which a model’s representation can still reorganize in response to new information [1, 2]. Our reformulation of FI for SSL leverages this property in a way that aligns with the role of pretraining: SSL defines the representation prior that downstream tasks inherit, and this prior is most effective when it remains broadly adaptable. Tracking FI during SSL pretraining therefore offers a qualitative signal of when this prior is still evolving and when it begins to lose adaptability.
>
> Integrating the additional elements suggested by the reviewer is a natural next step, and we plan to pursue this direction in future work.
>
> [1] Achille, Alessandro, Matteo Rovere, and Stefano Soatto. "Critical learning periods in deep networks." ICLR. 2018.
>
> [2] Berariu, Tudor, et al. "A study on the plasticity of neural networks." arXiv. 2021.
>
> > **W2 Need to provide more insights about why the FI dynamic performance differently among different SSL architectures. Are these differences come from the loss term or the contrastive and reconstruction-based SSL methods?**
>
> We have expanded our explanation of why SimCLR behaves differently (updated in Sec. 4.1). In brief, SimCLR is the only method in our comparison that must continuously balance two competing forces: SimCLR pulls together augmented views of the same image, and at the same time pushes that representation away from every other sample in the batch. This negative-pair pressure does not stabilize quickly. We attribute the longer critical period regime to the ongoing need to reorganize many sample-to-sample relationships. In contrast, DINO, VICReg, and MAE do not rely on this broad set of pairwise comparisons.
>
> To better understand why the critical period differs across objectives, a useful future direction is to examine how the representation grows and stabilizes during pretraining. Prior work shows that SSL methods often expand their representation dimensionality in a stepwise fashion [1]. Tracking how quickly each objective activates and stabilizes new directions in the representation, for example through the embedding covariance spectrum [2, 3], could clarify why some methods reach a stable configuration sooner than others and therefore close their critical periods at different times.
>
> [1] Simon, James B., et al. "On the stepwise nature of self-supervised learning." ICML. 2023.
>
> [2] Ziyin, Liu, et al. "What shapes the loss landscape of self supervised learning?." ICLR. 2023.
>
> [3] Agrawal, Kumar K., et al. "$\alpha $-ReQ: Assessing Representation Quality in Self-Supervised Learning by measuring eigenspectrum decay." NeurIPS. 2022.
>
> > **Q1 Can Fisher Information be replaced with a more stable unsupervised proxy?**
>
> To the best of our knowledge, Fisher Information (FI) is the most widely used proxy for information plasticity in neural networks [1, 2], and it has been empirically linked to generalization dynamics in early training [3]. That said, recent unsupervised representation metrics such as RankMe [4] and LiDAR [5] can be explored as complementary signals. These methods quantify structure in the learned representations, for example by measuring how spread out the feature dimensions are or by emphasizing feature directions that are consistently preserved across augmented views. However, they do not directly capture the plasticity properties we aim to study. Combining FI with representation-geometry metrics (e.g., RankMe, LiDAR) can shed deeper insight into how SSL representations evolve over time. We acknowledge these directions in our updated future work.
>
> [1] Achille, Alessandro, Matteo Rovere, and Stefano Soatto. "Critical learning periods in deep networks." ICLR. 2018.
>
> [2] Berariu, Tudor, et al. "A study on the plasticity of neural networks." arXiv. 2021.
>
> [3] Jastrzebski, Stanislaw, et al. "Catastrophic fisher explosion: Early phase fisher matrix impacts generalization." ICML. 2021.
>
> [4] Garrido, Quentin, et al. "Rankme: Assessing the downstream performance of pretrained self-supervised representations by their rank." ICML. 2023.
>
> [5] Thilak, Vimal, et al. "Lidar: Sensing linear probing performance in joint embedding ssl architectures." ICLR. 2024.

---

> > ### Author Response · Authors · 2025-11-28
> >
> > > **Q2 Do critical periods also exist in language or multi-modal SSL?**
> >
> > The reviewer raises a timely question. Our work focused on vision-based SSL, and we agree that extending critical period (CP) analysis to language and multimodal settings is a valuable direction for future work (denoted in Sec. 5).
> >
> > As a preliminary exploration, we examined a CLIP-style model for geo-foundation models (e.g., SatCLIP [1]) and observed a trend similar to our main findings: earlier checkpoints around 100 epochs (CP closure) outperform the original 500-epoch SatCLIP model on several downstream tasks (updated in Fig. 17, Appendix E).
> >
> > Although this single case is not sufficient to support general claims about multimodal SSL, it suggests that CP-like behavior may not be limited to unimodal vision models.
> >
> > [1] Klemmer, Konstantin, et al. "Satclip: Global, general-purpose location embeddings with satellite imagery." AAAI. 2025.
> >
> > > **Q3 How the model capacity will influence the CP closure?**
> >
> > To provide an initial check, we compared a ViT-Small and ViT-Base under identical DINO pretraining and found that, while both models exhibited CP closure, the larger model’s CP closed more slowly (updated in Fig. 18, Appendix E). This behavior is expected: larger models have higher representational capacity [1, 2, 3], which allows them to explore a broader feature space during pretraining and consequently delays the consolidation of their representations.
> >
> > These results suggest that model size shifts the timing of CP closure but not the existence of CP behavior. Extending this analysis to billion-scale models would definitely be an exciting direction for future work.
> >
> > [1] Szegedy, Christian, et al. "Going deeper with convolutions." CVPR. 2015.
> >
> > [2] He, Kaiming, et al. "Deep residual learning for image recognition." CVPR. 2016.
> >
> > [3] Touvron, Hugo, et al. "Three things everyone should know about vision transformers." ECCV. 2022.

---

> > > ### Comment · Reviewer_EgnN · 2025-11-28
> > >
> > > Thank you for your response. All of my concerns have been addressed. I will maintain my evaluation as weak accept but with a higher confidence

---

### Official Review · Reviewer_2iUr · 2025-10-30

**Soundness:** 3
**Presentation:** 3
**Contribution:** 2
**Rating:** 6
**Confidence:** 4

**Summary:**

The paper studies how self-supervised learners evolve over training and argues that there exist “critical periods” (CP) during which representations are highly plastic and best for transfer. The authors (i) document a transferability trade-off: longer pretraining continues to improve in-domain accuracy while hurting out-of-domain generalization, (ii) operationalize CP analysis for SSL via two tools, and (iii) propose two practical mechanisms: CP-guided checkpoint selection and CP-guided self-distillation that transfers intermediate “sweet-spot” layer features into the final model to balance ID and OOD performance.

**Strengths:**

1. The idea of rethinking critical periods without labels by analyzing deficits and Fisher Information on the pretext loss is conceptually elegant and broadly applicable.
2. The CP-guided checkpoint selection provides a simple yet practical tuning mechanism. The selective self-distillation strategy offers a concrete approach to restore transferability while retaining late-stage in-domain performance gains.

**Weaknesses:**

1. The paper directly equates Fisher Information on the self-supervised pretext objective with parameter sensitivity to supervision signals to quantify plasticity, but it does not establish a theoretical connection between this proxy and transfer performance or generalization error.
2. The paper posits a trade-off between sustained in-domain improvement and degraded out-of-domain transferability, which motivates the critical-period perspective, but lacks experiments linking this phenomenon to representation drift or task specialization.
3. The use of broad phrases such as “across datasets or distributions” suggests generality, yet the paper neither defines nor categorizes the types of distribution shifts considered, nor decomposes which shift types benefit most, making the scope of conclusions unclear.
4. The abstract claims coverage “across multiple SSL methods, architectures, and datasets,” but omits explicit statements on systematic coverage and boundary conditions. Without clarifying the circumstances under which the findings fail, the general conclusions risk overextension.

**Questions:**

1. Why do different SSL methods in Section 2.2 exhibit distinct trends, and why do models such as SimCLR not show a clear “critical period” phase?
2. How is Fisher Information computed in practice? Do you use full, block-diagonal, or diagonal approximations, per layer or per parameter group, and how frequently is it recomputed during pretraining?

---

> ### Author Response · Authors · 2025-11-28
>
> > **W1 The paper directly equates Fisher Information on the self-supervised pretext objective with parameter sensitivity to supervision signals to quantify plasticity, but it does not establish a theoretical connection between this proxy and transfer performance or generalization error.**
>
> We thank the reviewer for raising this point. We agree that Fisher Information (FI) is a proxy rather than a theoretically derived predictor of downstream transfer performance. We use FI because it provides a way to monitor when the model remains sensitive to its pretext supervision signal during SSL pretraining.
>
> This choice is supported by prior work in supervised learning showing that FI reflects changes in a network’s learning dynamics. [1] shows that FI rises sharply early in training and then declines, a pattern interpreted as a loss of “information plasticity,” meaning the network’s ability to reorganize its internal representations when exposed to new information. [2] further shows that early-phase FI dynamics empirically correlate with generalization. While these findings do not establish a theoretical guarantee, they suggest that changes in FI over time offer a useful indication of when a model remains adaptable versus when its representations have begun to stabilize.
>
> Since SSL pretraining defines the initial representation prior that later supports a suite of downstream tasks, tracking FI in our setting helps identify when this prior is still evolving and when it starts to become less adaptable.
>
> [1] Achille, Alessandro, Matteo Rovere, and Stefano Soatto. "Critical learning periods in deep networks." ICLR. 2018.
>
> [2] Jastrzebski, Stanislaw, et al. "Catastrophic fisher explosion: Early phase fisher matrix impacts generalization." ICML. 2021.
>
> > **W2 The paper posits a trade-off between sustained in-domain improvement and degraded out-of-domain transferability, which motivates the critical period perspective, but lacks experiments linking this phenomenon to representation drift or task specialization.**
>
> This is a very relevant point, and we added experiments measuring representation drift across checkpoints in two settings: 1) drift between consecutive checkpoints and 2) drift relative to an initial reference checkpoint (updated in Fig. 16, Appendix E). For a fixed set of validation images, we compute the mean feature-space distance between representations extracted by two checkpoints, which provides an estimate of how much the representation changes as training progresses [1].
>
> We find that drift is large early in pretraining (e.g., 6.22 at epochs 50→100) and decreases steadily as pretraining progresses (reaching 0.57 at epochs 950→1000). When measured relative to a fixed early reference (epoch 50), drift grows rapidly and then stabilizes. This pattern partially reinforces our observation that early phases exhibit representational plasticity, which may relate to improved downstream adaptation.
>
> [1] Pashakhanloo, Farhad, and Alexei Koulakov. "Stochastic gradient descent-induced drift of representation in a two-layer neural network." International conference on machine learning. PMLR, 2023.
>
> > **W3 The use of broad phrases such as “across datasets or distributions” suggests generality, yet the paper neither defines nor categorizes the types of distribution shifts considered, nor decomposes which shift types benefit most, making the scope of conclusions unclear.**
>
> We thank the reviewer for the observation. In the revised manuscript, we have replaced broad phrases such as “across datasets or distributions” to avoid unintended implications of generality. We clarify the scope and limitations of our analysis (updated in Sec. 2.1 and Sec. 5).
>
> > **W4 The abstract claims coverage “across multiple SSL methods, architectures, and datasets,” but omits explicit statements on systematic coverage and boundary conditions. Without clarifying the circumstances under which the findings fail, the general conclusions risk overextension.**
>
> Thank you for the clarification point. We have updated the manuscript to note that our conclusions are grounded in the SSL settings we evaluate, rather than implying generalizability to all possible settings (updated in Sec. 2.1 and Sec. 5).

---

> > ### Author Response · Authors · 2025-11-28
> >
> > > **Q1 Why do different SSL methods in Section 2.2 exhibit distinct trends, and why do models such as SimCLR not show a clear “critical period” phase?**
> >
> > We have expanded our explanation of why SimCLR behaves differently (updated in Sec. 4.1). In brief, SimCLR is the only method in our comparison that must continuously balance two competing forces: SimCLR pulls together augmented views of the same image, and at the same time pushes that representation away from every other sample in the batch. This negative-pair pressure does not stabilize quickly. We attribute the longer critical period regime to the ongoing need to reorganize many sample-to-sample relationships. In contrast, DINO, VICReg, and MAE do not rely on this broad set of pairwise comparisons.
> >
> > To better understand why the critical period differs across objectives, a useful future direction is to examine how the representation grows and stabilizes during pretraining. Prior work shows that SSL methods often expand their representation dimensionality in a stepwise fashion [1]. Tracking how quickly each objective activates and stabilizes new directions in the representation, for example through the embedding covariance spectrum [2, 3], could clarify why some methods reach a stable configuration sooner than others and therefore close their critical periods at different times.
> >
> > [1] Simon, James B., et al. "On the stepwise nature of self-supervised learning." ICML. 2023.
> >
> > [2] Ziyin, Liu, et al. "What shapes the loss landscape of self supervised learning?." ICLR. 2023.
> >
> > [3] Agrawal, Kumar K., et al. "$\alpha $-ReQ: Assessing Representation Quality in Self-Supervised Learning by measuring eigenspectrum decay." NeurIPS. 2022.
> >
> > > **Q2 How is Fisher Information computed in practice? Do you use full, block-diagonal, or diagonal approximations, per layer or per parameter group, and how frequently is it recomputed during pretraining?**
> >
> > We compute a diagonal Fisher approximation, as done in prior works [1, 2]. During pretraining, we compute the Fisher trace at every epoch, which provides a consistent view of how overall parameter sensitivity evolves over time.
> >
> > [1] Achille, Alessandro, Matteo Rovere, and Stefano Soatto. "Critical learning periods in deep networks." ICLR. 2018.
> >
> > [2] Kim, Youngeun, et al. "Exploring temporal information dynamics in spiking neural networks." AAAI. 2023.

---

> > > ### Comment · Reviewer_2iUr · 2025-11-28
> > >
> > > Thanks for your detailed response. All my concerns are addressed, and I'll maintain the positive score.

---

### Official Review · Reviewer_HAzr · 2025-10-30

**Soundness:** 3
**Presentation:** 4
**Contribution:** 3
**Rating:** 8
**Confidence:** 4

**Summary:**

This paper studies the question of how long to train a visual self-supervised learning model, by looking at the critical periods happening throughout training. The paper presents metrics that allow to identify these periods, and studies how they correlate to downstream tasks. Finally, from these metrics the paper describes methods to reconcile optimal in-domain and out-of-domain performance on downstream tasks. Experiments are conducted on the ImageNet-1K and fMoW-RGB satellite imagery datasets, and several popular SSL methods are studied (SimCLR, VICReg, DINO and MAE).

**Strengths:**

- The paper highlights a very interesting phenomenon in SSL, the fact that there exists learning phases that correlate with different levels of performance in out-of-domain or in-domain tasks.  The paper calls this phenomenon, which is the counterpart of overfitting in supervised learning, but for self-supervised learning: “critical periods”. The paper clearly explains how they found these critical periods with precise metrics, how these metrics correlate to downstream tasks and how to leverage these metrics to derive SSL models with better generalization capabilities.

- There is existing large-scale evidence of this phenomenon in the SSL literature. Most SSL practitioners have already encountered these critical phases without putting a name on it.

- The paper is well-written, easy to follow and with good presentation, the message and finding is simple and presented with clear experiments.

**Weaknesses:**

- The paper focuses on two datasets: ImageNet, which make sense as a general pretraining datasets to learn visual representation, and fMoW-RGB, a satellite imagery datasets, so not generalist. But the use fMoW-RGB is not well motivated and does not help make the experiment convincing. Is the motivation to clearly identify what is ID and what is OOD ? I would appreciate more a focus on ImageNet data or even larger scale generalist data, with the objective to see if these critical periods are also observed in real-world or large-scale scenarios. There is some evidence in the literature that this is the case, in DINOv3, they use gram anchoring at the end of their pretraining to retrieve pic segmentation performance that the model has towards the beginning of training, this is very similar to what the authors describe line 78 “Intermediate checkpoints often achieve better out-of-domain (OOD) transfer than later checkpoints”.

- The paper does not study critical phases with the prism of data overfitting, or doing too many passes on the same data. What would happen in an infinite data regime ? Would we still observe the same behaviour ? It would be great to have more experiments controlling the data distribution and studying the impact on these critical periods.

- Other than the impact of data, the paper also misses the fact that a schedule is used on the learning rate and weight decay. Both are brought to 0 progressively throughout the training and that could be correlated with the critical period observed. I think some experiments with a fixed schedule should be conducted to remove this confounding factor.

- Other similar metrics that have the same objective of characterizing SSL features are not acknowledged. For example, RankMe or LiDAR.

- The introduction should say a little more about the experimental setup, in terms of dataset used and concrete results obtained.

**Questions:**

Is there a link between critical periods and double descent ?

---

> ### Author Response · Authors · 2025-11-28
>
> > **W1 The paper focuses on two datasets: ImageNet, which make sense as a general pretraining datasets to learn visual representation, and fMoW-RGB, a satellite imagery datasets, so not generalist.**
>
> We thank the reviewer for the helpful comments. We have further emphasized the ImageNet results in the main paper. Regarding fMoW, our motivation is twofold. First, as the reviewer notes, it provides a clear setting for studying OOD behavior: the dataset contains natural geographic domain shifts that cleanly separate ID and OOD cases without synthetic manipulation. In particular, fMoW-WILDS [1] formalizes these real distribution shifts across space and time and is widely used for benchmarking OOD robustness. Second, fMoW is also a large-scale, real-world dataset (over one million images) that is used to pretrain foundation models for real-world applications. Including fMoW therefore allows us to test whether the critical period behavior persists beyond object-centric natural imagery and into a visually distinct, large-scale domain where OOD shifts are intrinsic to the data. We have clarified this motivation in the updated manuscript. Nonetheless, exploring even larger pretraining datasets, including multimodal data, is a natural next step, and we consider this an exciting avenue for future work.
>
> [1] Koh, Pang Wei, et al. "Wilds: A benchmark of in-the-wild distribution shifts." ICLR. 2021.
>
> > **W2 The paper does not study critical phases with the prism of data overfitting, or doing too many passes on the same data. ... It would be great to have more experiments controlling the data distribution and studying the impact on these critical periods.**
>
> We thank the reviewer for the interesting perspective. We agree that disentangling critical periods through the lens of data overfitting is a valuable direction. To approximately test this, we added a simple toy experiment that varies how many times each minibatch is reused before a single optimizer update (updated in Fig. 15, Appendix E). Reusing a batch multiple times forces the model to take repeated gradient steps on identical examples, increasing the degree of data reuse and overfitting pressure, whereas using each batch only once minimizes this effect.
>
> We observe that increasing data reuse shifts the onset of the critical period earlier, but the qualitative rise-peak-stabilize pattern exists. Exploring additional controlled data distribution manipulations is an interesting direction for future work, and we thank the reviewer for the suggestion.
>
> > **W3 Other than the impact of data, the paper also misses the fact that a schedule is used on the learning rate and weight decay. ... I think some experiments with a fixed schedule should be conducted to remove this confounding factor.**
>
> Thank you for this comment. We conducted an additional experiment using a fixed learning rate (LR) during pretraining (updated in Fig. 13, Appendix E). We find that the temporal sensitivity pattern is essentially unchanged: early deficits still produce the largest degradation, whereas later deficits have progressively smaller effects. This indicates that the observed phase-like behavior is not solely an artifact of the LR schedules.
>
> > **W4 Other similar metrics that have the same objective of characterizing SSL features are not acknowledged. For example, RankMe or LiDAR.**
>
> We appreciate the pointers to RankMe [1] and LiDAR [2]. These metrics assess representation quality in a label-free manner and are naturally complementary to our critical period (CP) analysis. They offer additional signals about representation structure that, together with our CP probes, can help clarify the temporal aspects of representation formation during SSL pretraining. We have added both references to the updated manuscript (Sec. 5).
>
> [1] Garrido, Quentin, et al. "Rankme: Assessing the downstream performance of pretrained self-supervised representations by their rank." ICML. 2023.
>
> [2] Thilak, Vimal, et al. "Lidar: Sensing linear probing performance in joint embedding ssl architectures." ICLR. 2024.
>
> > **W5 The introduction should say a little more about the experimental setup, in terms of dataset used and concrete results obtained.**
>
> We have updated the introduction to include a clearer description of the experimental setup.

---

> > ### Author Response · Authors · 2025-11-28
> >
> > > **Q1 Is there a link between critical periods and double descent?**
> >
> > This is a very interesting question. According to [1], double descent (epoch-wise) refers to the non-monotonic test error pattern as training progresses: test error initially falls, then rises to a peak, and falls again with continued training. This happens because longer training increases the model’s effective complexity [1], pushing it through a regime where generalization worsens before improving again.
> >
> > In terms of downstream transfer performance, this would correspond to performance improving, then sharply degrading, and then improving again. In our SSL setting, we do not clearly observe this behavior. However, both phenomena involve non-monotonic changes, and investigating whether a deeper relationship could exist is a promising direction.
> >
> > [1] Nakkiran, Preetum, et al. "Deep double descent: Where bigger models and more data hurt." ICLR. 2020.

---

### Official Review · Reviewer_12fS · 2025-10-31

**Soundness:** 3
**Presentation:** 3
**Contribution:** 4
**Rating:** 8
**Confidence:** 4

**Summary:**

This paper studies the training dynamics of self-supervised learning methods and how performance evolves both in domain and out of domain during training. To understand why OOD performance drops even though ID performance increases during training the authors people to study this behavior through the lens of critical periods. But using two different criteria, the authors are effectively able to detect when overspecialization starts to happen which helps select the best model for OOD performance, or a more balanced model.
Finally, the authors motivate and experiment with a distillation technique to help with overspecialization.

**Strengths:**

- The authors provide clear evidence of the studied problem, notably the drop in OOD performance during longer training

- The proposed metrics, in particular the one based on Fisher Information, correlate really well with this behavior. This leads to an effective method to perform early-stopping

- Experiments are performed at a good scale (R50/ViT-B, trained up to ImageNet for 1000 epochs) which adds to the relevance of the results

- The proposed distillation method is useful and very relevant to current SSL research. A similar problem was shown to be present in DINOv3[1], which concurrently proposed another solution based on earlier checkpoint distillation.

[1] Siméoni, Oriane, et al. "Dinov3." arXiv preprint arXiv:2508.10104 (2025).

**Weaknesses:**

1) Focus on Satellite dataset in the main paper. The same results as in the main paper are performed on ImageNet in the appendix but should be emphasised more in the main paper to appeal to a broader audience.

2) Throughout the paper, the considered evaluation is finetuning. However, the considered methods are more commonly used with lighter evaluations such as training a linear classifier. This would help make the results more relevant and may shed different insights.

**Questions:**

1) For Probe 1, how much do you think that the sensitivity is correlated with a lower learning rate later in training ?
2) Lines 254-255: when using noise during the deficit window, are data augmentation (jitter,crop,masking etc) still applied ? If not, how are the input image pairs constructed ?
3) Figure 4: All methods except VICReg have their Fisher Information drop to zero, do you have any intuition why it does not for VICReg ?

---

> ### Author Response · Authors · 2025-11-28
>
> > **W1 Focus on Satellite dataset in the main paper. The same results as in the main paper are performed on ImageNet in the appendix but should be emphasised more in the main paper to appeal to a broader audience.**
>
> Thank you for the helpful pointer. We have added additional description and context to highlight the ImageNet-based results more clearly.
>
> > **W2 Throughout the paper, the considered evaluation is finetuning. However, the considered methods are more commonly used with lighter evaluations such as training a linear classifier. This would help make the results more relevant and may shed different insights.**
>
> In line with the reviewer's feedback, we additionally ran linear probing evaluations at multiple checkpoints updated in Fig. 14 (Appendix E). The linear-probe results follow the same overall pattern as the finetuned results, though the overall trade-off tends to appear slightly later. This is expected because linear probing freezes the encoder and trains only a linear classifier on top. The classifier can therefore use only the information that is already linearly decodable at that checkpoint. As a result, we expect linear probing to require more pretraining before the relevant features become usable by the linear classifier.
>
> Regardless, the trend is consistent: intermediate checkpoints still yield the strongest OOD transfer, and continued pretraining past the critical period leads to overspecialization.
>
> > **Q1 For Probe 1, how much do you think that the sensitivity is correlated with a lower learning rate later in training?**
>
> We appreciate the reviewer’s question regarding the potential influence of the decaying learning rate on the Probe 1 sensitivity patterns. To remove any confounding factors, we conducted an additional experiment using a fixed learning rate throughout pretraining while keeping all other settings identical (updated in Fig. 13, Appendix E).
>
> We find that the temporal sensitivity pattern is essentially unchanged: early deficits still produce the largest degradation, whereas later deficits have progressively smaller effects. This indicates that the observed deficit-induced critical period is not an artifact of the learning rate schedule.
>
> > **Q2 Lines 254-255: when using noise during the deficit window, are data augmentation (jitter, crop, masking etc) still applied? If not, how are the input image pairs constructed?**
>
> Another great question. During the deficit window, we keep the full SSL augmentation pipeline unchanged and inject the deficit noise after the augmented views are generated. In other words, each method constructs its input views exactly as usual, and the deficit acts only as an additional perturbation applied on top of these augmented inputs. This preserves the method-specific view construction process while allowing the deficit to perturb the pretext signal directly.
>
> > **Q3 Figure 4: All methods except VICReg have their Fisher Information drop to zero, do you have any intuition why it does not for VICReg?**
>
> The reviewer makes an excellent observation. In our analysis, we focus primarily on the qualitative shape of the Fisher Information dynamics across SSL methods. As shown in the plots, VICReg displays the same rise, peak, and stabilization pattern as the other SSL methods, although as the reviewer notes its Fisher Information does not plateau like others. While we have not yet conducted thorough experiments, we hypothesize that the variance term in VICReg introduces a decorrelation mechanism driven by redundancy reduction and covariance regularization, which somehow influences the trend we observe. In future work, we intend to incorporate VICReg’s variance term into other SSL methods to obtain a definitive conclusion.

---

> > ### Comment · Reviewer_12fS · 2025-11-28
> >
> > Thank you for the answers that have helped clarify the scope of the study and made some results/experimental settings clearer. I will keep my already positive score.

---

### Official Review · Reviewer_U7ZN · 2025-11-06

**Soundness:** 4
**Presentation:** 4
**Contribution:** 4
**Rating:** 6
**Confidence:** 4

**Summary:**

This paper aims to answer the following question: "how long should SSL models be pretrained?". They relate this problem to the notion of critical periods: where models exhibit high plasticity in early training stages, and go through a consolidation phase where OOD adaptability declines but ID performance improves. This phenomenon has been observed in supervised learning and this work claims to be the first systematic investigation of critical periods in SSL. They make three main contributions:
- how to do CP analyses for SSL without requiring labels
- how CP closure can guide the process of selecting intermediate checkpoints that show stronger OOD robustness
- propose a distillation technique that uses sweet spot CP checkpoints as teacher and overspecialized networks as student, which leads to improved OOD generalization while maintaining ID performance.

**Strengths:**

1. **Reformulation of CP for SSL**: This seems to be the main novelty of this work. To show critical periods in SSL, the authors have proposed two analyses techniques: (a) perturbations at different learning stages (b) FI matrix with respect to pretext tasks.
2. **Practical Impact**: Contributions in Section 4 (CPCS and CPSD) can potentially have good practical impact providing guidance to select checkpoints based on transferability tradeoff, and improving OOD robustness.
3. **Experiments**: This work is backed well with strong experiments. They have considered two real-world datasets (IM-1K and fMoW-rgb) and four methods (SimCLR, VICReg, MAE, DINO). There're some DINOv2 results as well but I am not sure why it's only mentioned in Appendix.
4. **Reproducibility**: Hyperparameters and dataset details are well documented, and the methodology seems reproducible given the provided information.
5. **Quality & Clarity**: Overall, the paper is well-written and has a logical flow. The schematic in Figure 1 is particularly effective in summarizing the conceptual framework, which is later supported by quantitative results.

**Weaknesses:**

1. **Lack of Mathematical Rigor / Theoretical Depth**: The study is primarily empirical. While the FI metric offers some analytical grounding, the paper does not provide a principled explanation for why SSL exhibits critical periods or overspecialization. Some lightweight theoretical reasoning could strengthen the argument.

2. **Issues with CP-guided self-distillation**:

    (a) L117: it is not explained why the authors chose to distill early layers only. It's only later in Section 4.3 where the rationale behind this choice is addressed.

    (b) Figure 6 is not convincing enough to justify distilling early layers. I think it will be useful to verify their claims by distilling the entire network and comparing performance gains.

    (c) **Missing Ablations**: The number of layers distilled (L) and the distillation weight (λ) are unspecified and unexplored. Both likely affect results and reproducibility.

3. **Figure 6**: It is not clear what the authors imply by "stage".
4. **SimCLR's behavior**: I appreciate the attempt to explain why SimCLR's critical period closes much later (L375-376). I think this section needs more clarification. What are the differences in objectives of all the methods taken into consideration that leads to this behavior?
5. **Definition of CP closure**: The paper briefly mentions detecting CP closure “L388- when the FI slope stabilizes (e.g., below a tolerance for p consecutive epochs)” (in Sec. 4.2). However, it is unclear whether this rule was actually used to determine the CP checkpoints reported in the experiments. For reproducibility and clarity, it would be valuable for the authors to specify the exact tolerance threshold and window length used in practice (if any).

**Questions:**

Please refer to weaknesses.

**Details Of Ethics Concerns:**

No concerns.

---

> ### Author Response · Authors · 2025-11-28
>
> > **W1 The study is primarily empirical. While the FI metric offers some analytical grounding, the paper does not provide a principled explanation for why SSL exhibits critical periods or overspecialization. Some lightweight theoretical reasoning could strengthen the argument.**
>
> The reviewer's observation is on-point, and we agree that providing some theoretical insights into why SSL may exhibit critical period-like behavior would strengthen the study and provide additional benefits. To shed light on this, we turn to insights from prior information-theoretic analyses.
>
> In the Information Bottleneck (IB) view [1], representation learning involves a tension between retaining the information in the input that is useful for the training objective and compressing away variability that is task-irrelevant [2, 3]. In SSL, the “task’’ that defines this objective is the \textit{pretext task}: it is the only source of supervision during SSL pretraining and therefore determines what information the model is incentivized to preserve.
>
> While we do not claim that SSL explicitly optimizes an IB objective, the IB view suggests an explanation for the critical period-like behavior we observe. Early in training, the representation is not yet constrained by the invariances or reconstruction patterns favored by the pretext task, so optimization has not begun to aggressively compress away input variability. As pretraining continues, the pretext objective increasingly shapes the representation, and the learning dynamics suppress aspects of the input that do not help reduce the pretext task. This creates a natural trade-off [4]: too little compression leaves the representation underdeveloped and not yet useful for downstream tasks, whereas too much compression can overspecialize the representation to the pretext objective and discard information that would be useful (transferable) for other domains.
>
> We have added a brief explanation in Appendix F to reflect this intuition while keeping our claims modest and empirical.
>
> [1] Shwartz-Ziv, Ravid, and Naftali Tishby. "Opening the black box of deep neural networks via information." arXiv. 2017.
>
> [2] Ouyang, Zhuo, et al. "Projection head is secretly an information bottleneck." ICLR. 2025.
>
> [3] Shwartz Ziv, Ravid, and Yann LeCun. "To compress or not to compress—self-supervised learning and information theory: A review." Entropy. 2024
>
> [4] Tian, Yonglong, et al. "What makes for good views for contrastive learning?." NeurIPS. 2020.
>
> > **W2(a) L117: it is not explained why the authors chose to distill early layers only. It's only later in Section 4.3 where the rationale behind this choice is addressed.**
>
> We have updated the manuscript to provide this intuition earlier in the introduction (lines 125–127). Briefly, we note that we distill early layers because they are generally understood to contain the more transferable features [1], which may degrade with prolonged pretraining as the model increasingly adapts to the training objective, whereas later layers tend to encode task-specific information [2].
>
> [1] Yosinski, Jason, et al. "How transferable are features in deep neural networks?." NeurIPS. 2024
>
> [2] Bordes, Florian, et al. "Guillotine regularization: Why removing layers is needed to improve generalization in self-supervised learning." TMLR. 2023.
>
> > **W2(b) Figure 6 is not convincing enough to justify distilling early layers. I think it will be useful to verify their claims by distilling the entire network and comparing performance gains.**
>
> We have conducted additional experiments that distill the entire network rather than only the early layers. As shown in the updated Table 1, full-layer distillation performs worse than early-layer distillation. We conjecture that full-layer distillation pulls the entire model back toward the CP checkpoint, restoring some transferability but also undoing the ID-specific refinements encoded in later layers.
>
> > **W2(c) The number of layers distilled (L) and the distillation weight ($\lambda$) are unspecified and unexplored. Both likely affect results and reproducibility.**
>
> Thank you for this clarifying point. For the main results, we distill features from the first residual block group of ResNet-50 (the conv2\_x group [1], referred to as “stage 1”), using a distillation weight of $\lambda = 0.5$ (denoted in Appendix D). We also report additional experiments varying the distillation weight $\lambda$ in Table 3 (Appendix E), and find that performance gains are stable.
>
> [1] He, Kaiming, et al. "Deep residual learning for image recognition." CVPR. 2016
>
> > **W3 Figure 6: It is not clear what the authors imply by "stage".**
>
> In Fig. 6, the term “stage” refers to the architectural stages of a ResNet-50 backbone [1], that is, the four sequential groups of residual blocks with increasing channel width (dubbed conv2, conv3, conv4, and conv5 from [1]).
>
> [1] He, Kaiming, et al. "Deep residual learning for image recognition." CVPR. 2016

---

> > ### Author Response · Authors · 2025-11-28
> >
> > > **W4 I appreciate the attempt to explain why SimCLR's critical period closes much later (L375-376). I think this section needs more clarification. What are the differences in objectives of all the methods taken into consideration that leads to this behavior?**
> >
> > Thank you for the thoughtful comment. We have expanded our explanation of why SimCLR behaves differently (updated in Sec. 4.1). In brief, SimCLR is the only method in our comparison that must continuously balance two competing forces: SimCLR pulls together augmented views of the same image, and at the same time pushes that representation away from every other sample in the batch. This negative-pair pressure does not stabilize quickly. We attribute the longer critical period regime to the ongoing need to reorganize many sample-to-sample relationships. In contrast, DINO, VICReg, and MAE do not rely on this broad set of pairwise comparisons.
> >
> > To better understand why the critical period differs across objectives, a useful future direction is to examine how the representation grows and stabilizes during pretraining. Prior work shows that SSL methods often expand their representation dimensionality in a stepwise fashion [1]. Tracking how quickly each objective activates and stabilizes new directions in the representation, for example through the embedding covariance spectrum [2, 3], could clarify why some methods reach a stable configuration sooner than others and therefore close their critical periods at different times.
> >
> > [1] Simon, James B., et al. "On the stepwise nature of self-supervised learning." ICML. 2023.
> >
> > [2] Ziyin, Liu, et al. "What shapes the loss landscape of self supervised learning?." ICLR. 2023.
> >
> > [3] Agrawal, Kumar K., et al. "$\alpha $-ReQ: Assessing Representation Quality in Self-Supervised Learning by measuring eigenspectrum decay." NeurIPS. 2022.
> >
> > > **W5 The paper briefly mentions detecting CP closure “L388- when the FI slope stabilizes (e.g., below a tolerance for p consecutive epochs)” (in Sec. 4.2). However, it is unclear whether this rule was actually used to determine the CP checkpoints reported in the experiments. For reproducibility and clarity, it would be valuable for the authors to specify the exact tolerance threshold and window length used in practice (if any).**
> >
> > The reviewer raises another valuable point. In all runs, we save checkpoints every 50 epochs up to epoch 1000. Since CP closure will not always occur exactly at a multiple of 50 epochs, we identify the CP checkpoint by locating the first point at which the FI curve enters its plateau and then selecting the nearest saved checkpoint.
> >
> > To complement this, we also ran an additional experiment in which we saved checkpoints more densely (every 10 epochs) in the region where the Fisher Information begins to stabilize (grey shading marks in Fig. 5). In this finer-resolution setting, the best-performing checkpoints existed around the plateau point. We have included the corresponding results in Fig. 12 (Appendix E).

---

### Author Response · Authors · 2025-11-28
**Overall Summary**

We thank all reviewers for their accurate summary and thoughtful feedback of our work. We are encouraged that the reviewers found the problem important and timely, and the characterization of critical periods in self-supervised learning (SSL) methods clear and convincing.

In response to the valuable feedback across reviewers, we have made several improvements to the manuscript, now posted (updates marked in blue). At a high level, we summarize the main clarifications we have incorporated:

- Reviewer U7ZN suggested adding theoretical intuition for why SSL exhibits critical periods and validating early-layer distillation by comparing it to full-network distillation. We address these points by introducing an information-theoretic motivation and reporting full-network distillation results, which show weaker performance than early-layer distillation.

- Reviewer 12fS noted that including linear probe evaluations and clarifying confounds such as learning rate decay would strengthen the work. We incorporate these additions, and both linear probe and fixed learning rate experiments support our main findings.

- Reviewer HAzr raised the importance of better motivating our dataset choices and clarifying how data regime and training schedules relate to the observed critical period behavior. We address these points by strengthening the dataset motivation in the main text and adding experiments that control data reuse and learning rate schedule, both of which show trends consistent with our main findings.

- Reviewer 2iUr asked for a clearer connection between Fisher dynamics and transfer learning, as well as precise framing of the scope our evaluated settings. We address these by explaining how quantifying parameter sensitivity during SSL pretraining via FI provides an intuitive signal for downstream transfer, and by specifying the distribution shifts and evaluation scenarios.

- Reviewer EgnN asked for clarification on why Fisher dynamics differ across SSL methods and whether critical periods also arise in multimodal settings. We address these by expanding our explanation of how different SSL pretext tasks may induce different rates of representation stabilization and by adding preliminary multimodal experiments that show similar trends.

We thank all reviewers once again for their constructive feedback.

---

### Meta-Review · Area_Chair_n8RW · 2025-12-23

**Summary:**

This paper provides a systematic analysis of training dynamics in self-supervised learning through the lens of critical periods. It shows that while in-distribution performance improves with longer pretraining, out-of-distribution performance often peaks early and then degrades due to overspecialization. The authors propose label-free criteria to identify critical period closure, enabling better checkpoint selection for OOD robustness. They further introduce a distillation strategy that transfers knowledge from critical-period “sweet spot” models to later checkpoints, effectively mitigating overspecialization and improving OOD generalization without sacrificing in-distribution performance.

**Reviewer Concerns:**

The following concerns have been addressed:

(1)  theoretical intuition for why SSL exhibits critical periods

(2) additional validation of early-layer distillation

(3) additional numerical experiment regarding linear prob and learning rate decay.

(4) Clarification of the connection between Fisher dynamics and transfer learning

**Reviewer Scores:**

Reviewer U7ZN gave a score of 6. The reviewer did not participate in the discussion but is likely to maintain the score.

Reviewer 12fS gave a score of 8. The reviewer participated in the discussion and indicated that they will maintain the score.

Reviewer HAzr gave a score of 8. The reviewer did not participate in the discussion but is likely to maintain the score.

Reviewer 2iUr gave a score of 6. The reviewer participated in the discussion and indicated that they will maintain the score.

Reviewer EgnN gave a score of 6. The reviewer participated in the discussion and indicated that they will maintain the score.

Overall, all reviewers provided positive evaluations, with an average score of 6.8.

---

### Decision · Program_Chairs · 2026-01-26

Accept (Poster)